# Molecular profiling of tissue biopsies reveals unique signatures associated with streptococcal necrotizing soft tissue infections

Robert Thänert [1,11], Andreas Itzek[1,11], Jörn Hoßmann[1], Domenica Hamisch[1], Martin Bruun Madsen[2], Ole Hyldegaard[3], Steinar Skrede[4,5], Trond Bruun [4], Anna Norrby-Teglund[6], INFECT study group, Eva Medina[7,11] & Dietmar H. Pieper[1,11]

Necrotizing soft tissue infections (NSTIs) are devastating infections caused by either a single pathogen, predominantly *Streptococcus pyogenes*, or by multiple bacterial species. A better understanding of the pathogenic mechanisms underlying these different NSTI types could facilitate faster diagnostic and more effective therapeutic strategies. Here, we integrate microbial community profiling with host and pathogen(s) transcriptional analysis in patient biopsies to dissect the pathophysiology of streptococcal and polymicrobial NSTIs. We observe that the pathogenicity of polymicrobial communities is mediated by synergistic interactions between community members, fueling a cycle of bacterial colonization and inflammatory tissue destruction. In *S. pyogenes* NSTIs, expression of specialized virulence factors underlies infection pathophysiology. Furthermore, we identify a strong interferon-related response specific to *S. pyogenes* NSTIs that could be exploited as a potential diagnostic biomarker. Our study provides insights into the pathophysiology of mono- and polymicrobial NSTIs and highlights the potential of host-derived signatures for microbial diagnosis of NSTIs.

[1] Microbial Interactions and Processes Research Group, Helmholtz Center for Infection Research, Braunschweig, Germany. [2] Department of Intensive Care, Copenhagen University Hospital, Rigshospitalet, Copenhagen, Denmark. [3] Department of Anaesthesia, Centre of Head and Orthopaedics, Copenhagen University Hospital, Rigshospitalet, Copenhagen, Denmark. [4] Department of Medicine, Haukeland University Hospital, Bergen, Norway. [5] Department of Clinical Science, University of Bergen, Bergen, Norway. [6] Center for Infectious Medicine, Karolinska Institutet, Karolinska University Hospital, Huddinge, Sweden. [7] Infection Immunity Research Group, Helmholtz Center for Infection Research, Braunschweig, Germany. [11]These authors contributed equally: Robert Thänert, Andreas Itzek, Eva Medina, Dietmar H. Pieper. A full list of consortium members appears at the end of the paper. Correspondence and requests for materials should be addressed to D.H.P. (email: dpi@helmholtz-hzi.de)

Necrotizing soft tissue infections (NSTIs) are life-threatening bacterial infections characterized by rapidly spreading necrosis of the skin and subcutaneous tissues[1]. An immunonocompromised status, caused for example by chemotherapy, can predispose patients for NSTIs and diabetic ulcers, penetrating trauma, breach of the epithelial barrier, cirrhosis, or minor non-penetrating traumas are risk factors for NSTIs[1]. Diagnosis and management of NSTIs can be challenging as initial signs and symptoms can be ambiguous[2]. Despite the advances of modern medicine, mortality associated with NSTIs remains as high as 30% and escalates up to 70% if the correct clinical diagnosis is delayed[3,4].

The microbial etiology of NSTIs can either be monomicrobial, caused by a single bacterial species, or polymicrobial, caused by diverse microorganisms. While *S. pyogenes* is the most common pathogen associated with monomicrobial NSTIs[1], other streptococcal species (*S. dysgalactiae*, *S. agalactiae*) and *Staphylococcus aureus* have also been reported to cause monomicrobial NSTIs[5,6]. Polymicrobial NSTIs are commonly associated with a mixture of aerobic and anaerobic bacteria[7]. Among these, Enterobacteriaceae, *Bacteroides* spp., *Porphyromonas* spp., *Prevotella* spp., *Peptostreptococcus* spp., and *Clostridium* spp. have been most frequently isolated from infected tissue[2,8]. Whereas polymicrobial NSTIs are usually observed in older patients, or in individuals with underlying comorbidities such as diabetes, monomicrobial NSTIs are more commonly associated with trauma, surgery, or intravenous drug use[1,9]. The reported relative incidence of mono- and polymicrobial NSTIs varies substantially according to the geography and the specific characteristics of the patient cohort[1].

The pathophysiology of monomicrobial NSTIs caused by *S. pyogenes* has been studied extensively and many of the virulence factors and toxins expressed by the bacterium to efficiently colonize the host tissue, escape the host immune defenses and rapidly spread to surrounding tissue have been very well characterized[10]. Neutralization of toxins by intravenous administration of human immunoglobulins has been proposed, in addition to surgical debriment and antimicrobial treatment, as an additional adjunct therapy to improve the outcome of NSTIs caused by *S. pyogenes*[11]. In contrast to *S. pyogenes* NSTIs, mechanistic studies addressing the pathogenic strategies and complex dynamics of bacterial communities in polymicrobial NSTIs are lacking. This may be due to the technical challenges associated with investigations of the composition, structure, and activity of polymicrobial communities. Similarly, knowledge on the host response to polymicrobial NSTIs is scarce and it is unknown if and how the host response in polymicrobial NSTIs differs from that during monomicrobial infections. A better understanding of these processes is, however, crucial as it could facilitate accurate identification of the infecting microorganism(s) and would enable to develop treatment strategies tailored toward the microbial etiology.

Currently, drastic surgical debridement of the affected tissue is essential for successful treatment of NSTIs and total or partial limb amputation may be required in cases of severe NSTIs with survivors facing substantial risk for long-term morbidity and reduced quality of life[12]. Besides rapid surgical intervention, antibiotic therapy constitutes the most important adjunct in NSTIs treatment[13]. Empiric antibiotic treatment commonly consists of a mixture of broad-spectrum ß-lactams and fluroquinolones with antimicrobials used to treat anaerobic infections, such as Clindamycin or Metronidazole[4]. Once the causative agent(s) has been identified, antibiotic treatment regimes are revised to more specifically target the bacteria present within the infected tissue. To date, clinical microbial diagnosis of NSTIs is mostly based on bacterial cultures of pre- and intraoperatively obtained tissue- and blood samples[4]. In clinical practice, one major challenge in treating severe NSTIs is the time required for diagnosis. Therefore, diagnostic tools for NSTIs

need to be re-evaluated since the time effectiveness of the therapeutic intervention will increase the chances of survival of the patient. Conceptually, shortening the diagnostic time by the implementation of next-generation molecular tools that enable simultaneous identification of multiple microbes directly from clinical samples as well as easy to measure surrogate biomarkers of microbial etiology could accelerate and improve the management of this acute life-threatening infections. Importantly, such markers should be invariantly associated with specific microbial signatures and robust to fluctuations to allow reliable diagnosis at any phase of the infection. The realization of these diagnostic tools is hampered by a lack of basic knowledge of the microbial diversity associated with NSTIs, their pathological mechanisms as well as the host response in real time during infection.

In this study, we integrate microbial community profiling using 16S rRNA sequencing with transcriptional analysis of host and microbe using dual RNA-sequencing (RNA-seq) in tissue biopsies from NSTI patients to unravel the molecular mechanisms underlying the pathophysiology of mono- and polymicrobial NSTIs. Our results demonstrate that, despite the similar clinical presentation of NSTIs, the pathophysiology of mono- and polymicrobial etiology differs significantly and that these differences can potentially be exploited for diagnostic purposes.

## Results

**Identification of bacteria present in NSTIs biopsies.** To identify the microorganisms involved in NSTIs, 16S rRNA gene sequencing was performed in tissue biopsies obtained from 148 patients, collected at five clinical sites (Table 1). Consistent with previous reports[8,14], NSTIs were either dominated by a single bacterial pathogen or associated with a diverse spectrum of phylotypes. Taking the alpha-diversity as a measure, 41 cases, which were dominated by a single phylotype (>85% relative abundance), showed a Gini–Simpson diversity index<0.25 (referred to as 'monomicrobial'), whereas in 54 cases no single phylotype accumulated more than one third of sequences (Fig. 1a, Supplementary Data 1, 2, and 3). The majority of monomicrobial NSTIs were associated with *S. pyogenes* (29 of 41 cases), but other pathogens rarely reported in the context of NSTIs, such as *Staphylococcus aureus*, *Streptococcus dysgalactiae*, and *Streptococcus agalactiae*, were recovered from some of those biopsies (Supplementary Figure 1). *Clostridium sensu stricto* strains, historically considered major causes of tissue necrosis[15], were only sparsely observed (Supplementary Data 1).

The composition of bacterial communities in polymicrobial NSTIs was highly variable, but *Prevotella* spp., *Porphyromonas* spp., *Parvimonas* spp., *Fusobacterium* spp., *Peptostreptococcus* spp., and *Bacteroides* spp. were most frequently observed in high abundances (Fig. 1a, Supplementary Data 1, Supplementary Figures 2–7). Surprisingly, we observed extreme differences in the species level diversity within these highly abundant bacterial genera. Whereas the *Bacteroides* spp. abundance was dominated by *B. fragilis*, which was previously characterized as a keystone pathogen[16] and accounted for 78.5% of the total genus abundance, no single key species was observed among the other genera, with seven species accounting for 80% of the *Prevotella* genus abundance (Supplementary Figure 1).

To characterize microbial interactions within NSTIs, we inferred bacterial co-occurrence patterns of bacterial genera using CoNet, an ensemble method of network analysis[17]. We observed significant negative associations between pathogens causing monomicrobial NSTIs such as *Streptococcus* spp. and the taxonomically diverse genera detected in polymicrobial NSTIs (Fig. 1b, Supplementary Data 4). Using divisive clustering of co-occurring genera (see Methods), highly interconnected clusters of bacterial genera

## Table 1 Clinical characteristics of patients cohort by infection type

| | Streptococcus (n = 73) | Staphylococcus (n = 5) | Bacteroides/ Escherichia (n = 7) | Polymicrobial (n = 45) | Other (n = 18) |
|---|---|---|---|---|---|
| Age | 58.9 (8.5) | 54.2 (14.9) | 63.4 (16.3) | 59 (13.1) | 58.7 (10.8) |
| *Gender* | | | | | |
| Female/male | 28/45 | 0/5 | 1/6 | 23/22 | 8/10 |
| *Outcome* | | | | | |
| Mortality ICU (%) | 6.9% | 0% | 14.3% | 11.1% | 16.7% |
| Mortality 1 month (%) | 11.0% | 0% | 28.6% | 13.3% | 27.8% |
| Mortality 3 months (%) | 13.7% | 0% | 28.6% | 15.6% | 27.8% |
| Mortality 1 year (%) | 20.5% | 0% | 42.9% | 26.7% | 33.3% |
| *Hospital* | | | | | |
| Rigshospitalet Copenhagen | 37/73 | 4/5 | 4/7 | 25/45 | 12/18 |
| Karolinska University Hospital | 10/73 | 1/5 | 1/7 | 11/45 | – |
| Sahlgrenska University Hospital | 9/73 | – | 1/7 | 4/45 | 3/18 |
| University of Bergen | 15/73 | – | 1/7 | 4/45 | 2/18 |
| Blekingesjukhuset Karlskrona | 2/73 | – | – | 1/45 | 1/18 |
| Time from admission to surgery (hours)[a] | 53 (114) | 52 (51) | 41 (90) | 33 (56) | 192 (504) |
| *Laboratory values* | | | | | |
| Hemoglobin (g/l)[b] | 9.48 (2.08) | 9.89 (1.85) | 7.64 (2.00) | 8.82 (1.50) | 8.20 (1.90) |
| White blood cells ($10^9$/l)[c] | 17.18 (7.92) | 20.34 (4.75) | 20.94 (11.42) | 18.10 (8.96) | 13.90 (7.54) |
| C-reactive protein (mg/l)[d] | 240 (123) | 183 (58) | 216 (84) | 254 (117) | 198 (116) |
| Creatinine (μmol/l)[e] | 165 (124) | 86 (8) | 157 (106) | 154 (110) | 151 (108) |
| Sodium (mmol/l) | 134 (5) | 140 (6) | 134 (7) | 136 (4) | 134 (4) |
| Glucose (mmol/l)[f] | 10.9 (6.7) | 9.1 (3.6) | 12.8 (8.3) | 12.4 (5.1) | 10.5 (4.4) |
| *Physiological values* | | | | | |
| Pulse (beats/minute)[g] | 112 (22) | 95 (19) | 135 (28) | 107 (21) | 107 (25) |
| Mean arterial pressure (mmHg)[g] | 60 (8) | 59 (4) | 63 (4) | 59 (8) | 57 (12) |
| LRINEC score[h] | 7.3 (2.7) | 6.0 (3.1) | 7.0 (3.4) | 7.5 (2.4) | 7.7 (3.1) |

Data are the mean (s.d.) from 148 patients. Laboratory and physiological values pertain to the first 24 hours after ICU admission. For hemoglobin and sodium, the lowest recorded values are indicated. For white blood cells, C-reactive protein, creatinine and glucose, the highest recorded values are indicated. For pulse, the highest value is indicated whereas for mean arterial blood pressure, the lowest value is indicated. The LRINEC score includes sub scores ranging from 0 to a maximum of four for six different blood samples (hemoglobin, white blood cells, C-reactive protein, creatinine, sodium, and glucose). Aggregated scores range from 0 to 13, with higher scores indicating higher risk of necrotizing fasciitis
*LRINEC* Laboratory Risk Indicator for Necrotising Fasciitis
[a]Including patients who were primarily admitted for other reasons. Data regarding exact time of initial hospital admission were missing for three patients in the streptococcal group, one patient in the staphylococcal group and one patient in the polymicrobial group
[b]Data for hemoglobin were missing for one patient in the streptococcal group
[c]Data for white blood cells were missing for one patient in the streptococcal group
[d]Data for CRP were missing for three patients in the streptococcal group and for two patients in the polymicrobial group
[e]Data for creatinine were missing for one patient in the streptococcal group
[f]Data for glucose were missing for three patients in the streptococcal group for one patient in the polymicrobial group
[g]Data for pulse and mean arterial pressure were missing for one patient in the polymicrobial group
[h]Data for the LRINEC score were missing for 13 patients in the streptococcal group, for seven patients in the polymicrobial group and for three patients in the others group

associated with polymicrobial NSTIs were identified (Fig. 1b, Supplementary Figure 8a), suggesting that these microorganisms may act in concert to establish NSTIs.

Consistent with clinical reports[18,19], the diversity and composition of the bacterial communities in the infected tissue were highly dependent on the affected anatomic location (Fig. 1c, Supplementary Table 1). Thus, the bacterial diversity in NSTIs of the upper and lower extremities was significantly lower than in NSTIs localized at the head/neck or anogenital region indicating a higher frequency of polymicrobial infections in the latter (Fig. 1c). Concomitantly, we observed significantly higher abundances of gastrointestinal *Bacteroides* spp. in anogenital infections than at the extremities, while the predominately oral genera *Prevotella*, *Porphyromonas,* and *Fusobacterium* exhibited significantly increased abundances in infections of the head/neck area (Table 2). These observations highlight a potential connection between bacterial community composition in NSTIs and natural niches-associated commensals.

**Pathogenicity of streptococcal and polymicrobial NSTIs differs**. To characterize the mechanisms of bacterial pathogenicity in mono- and polymicrobial NSTIs, we performed simultaneous transcriptional profiling of infected human tissue and bacterial gene expression analysis via dual RNA-seq. Using hierarchical agglomerative clustering we classified patients in different clusters according to the pathogen(s) and microbial community composition associated with the obtained tissue biopsies (Fig. 2, Supplementary Data 3). Mortality rates did not differ between patients with either polymicrobial or streptococcal NSTIs

(Table 1, Supplementary Figure 8b). As the frequency of monomicrobial NSTIs caused by bacteria other than *Streptococcus* spp. was too low to enable statistical analysis, we randomly selected a set of monomicrobial *Streptococcus* spp. NSTI biopsies (n = 17) as well as a set of polymicrobial NSTIs (n = 22) for in-depth transcriptional profiling (Supplementary Data 3 and Fig. 2). A strong correlation between the relative contribution of identified bacterial genera to the transcriptional activity and the relative abundance of these genera within the NSTIs microbial communities as identified by 16S rRNA gene sequencing was observed (Supplementary Figure 9a, b). The bacterial transcriptional profile indicated expression of a broader spectrum of functionalities in polymicrobial than in monomicrobial NSTIs (Fig. 3, Supplementary Data 5). Analysis of highly transcribed functionalities indicated that specific pathways, such as purine ribonucleoside salvage or glycine catabolism, were only expressed by a fraction of the polymicrobial community, as evidenced by low associated alpha-diversity of the genus-level transcriptional contribution to these functionalities (Supplementary Figure 9c). This observation may indicate functional specialization of the co-occurring genera within the bacterial communities of polymicrobial NSTIs. Comparative analysis of streptococcal versus polymicrobial NSTIs indicated that carbohydrate metabolic pathways including lactose (GO:0005990), trehalose (GO:0005993), and galactose metabolism (Supplementary Figure 10) and carbohydrate transport (GO:0015771, GO:0008645, GO:0034219) were strongly expressed in monomicrobial NSTIs, while amino acid transport (GO:0006865, GO:0015813) and metabolism (Supplementary Figure 10), and protein processing

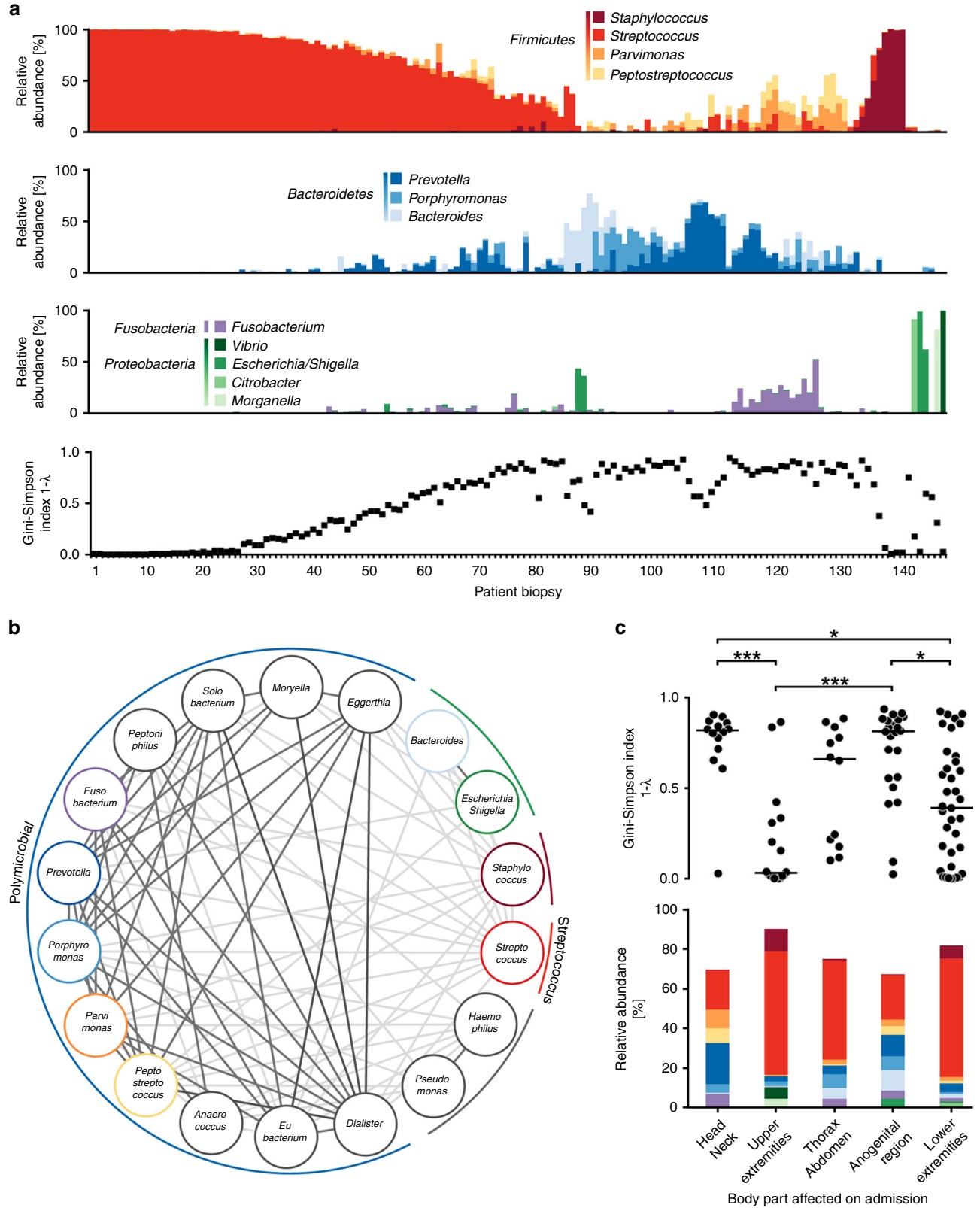

(GO:0016485) predominated in the transcriptional profile of the bacteria associated with polymicrobial NSTIs (Fig. 3, Supplementary Data 5). Furthermore, lipopolysaccharide core region biosynthesis (GO:0009244), lipid A biosynthesis (GO:0009245), and polysaccharide transport (GO:0015774), all of which are involved in the synthesis and transport of lipopolysaccharide

(LPS) were significantly enriched in the functional profile of polymicrobial NSTIs (Fig. 3, Supplementary Data 5). This expression of LPS, the integral component of the outer membrane of Gram-negative bacteria and powerful activator of a robust inflammatory response, may contribute to the inflammatory response observed in polymicrobial NSTIs.

**Fig. 1** Single pathogens as well as complex bacterial communities can cause severe NSTIs. **a** Bacterial composition in tissue biopsies ($n = 148$) from patients with NSTIs. Bacterial genera with a mean relative abundance of ≥2.5% across all samples or a maximal relative abundance of ≥80% are depicted. The Gini–Simpson diversity index shows genus-level diversity. **b** Co-occurrence network of the 18 genera with the highest mean relative abundance across all samples. Dark edges illustrate co-occurrence, light edges mutual exclusion (Brown's $p$-value ≤0.05). Outer lines represent distinct bacterial clusters (see Supplementary Figure 8). **c** Bacterial community diversity and structure in NSTI biopsies depicted against the affected body part. The bacterial community diversity is given to the top where each dot represents the Gini–Simpson diversity index from one specimen. Lines represent median values. Diversity indices across the five body parts ($n = 14, 18, 12, 23$, and 37, respectively from left to right) were compared using the Kruskal–Wallis test with Dunn's multiple comparison post hoc test with ***$p < 0.001$; *$p < 0.05$. The bacterial community structure is indicated to the bottom where bars depict the mean relative abundance of genera at each body site. The color code depicting the different genera is as given in **a**. For statistical evaluation of the relative abundance of genera at different body sites see Table 2. Source data are provided in Supplementary Data 1, 2, 3, and 4

### Table 2 Genera are differently abundant at affected body sites

|  | U/L | U/H | U/A | U/T | L/H | L/A | L/T | A/H | T/H | A/T |
|---|---|---|---|---|---|---|---|---|---|---|
| Streptococcus | ns | ns | ns | ns | ns | **0.0128** | ns | ns | ns | ns |
| Bacteroides | ns | ns | 0.0018 | ns | ns | 0.0308 | ns | ns | ns | ns |
| Parvimonas | ns | 0.0004 | ns | **0.0352** | 0.0071 | ns | ns | ns | ns | ns |
| Prevotella | ns | <0.0001 | 0.0038 | ns | 0.0029 | ns | ns | ns | ns | ns |
| Porphyromonas | ns | 0.0022 | 0.0052 | ns | 0.0271 | ns | ns | ns | ns | ns |
| Peptostreptococcus | ns | 0.0141 | ns | ns | 0.0093 | ns | ns | ns | ns | ns |
| Escherichia/Shigella | ns | ns | ns | 0.0292 | ns | ns | ns | ns | ns | ns |
| Fusobacterium | ns | 0.0015 | ns | ns | 0.0245 | ns | ns | ns | ns | ns |
| Eggerthia | ns | 0.0164 | ns | ns | 0.0039 | ns | ns | 0.0479 | ns | ns |
| Eubacterium | ns | 0.0016 | ns | ns | ns | ns | ns | 0.0435 | ns | ns |
| Solobacterium | ns | 0.0001 | ns | 0.0033 | 0.0045 | ns | ns | 0.0141 | ns | ns |
| Dialister | ns | 0.0002 | 0.0044 | ns | 0.0011 | 0.0257 | ns | ns | ns | ns |
| Bulleidia | ns | 0.0022 | ns | ns | 0.0171 | ns | ns | 0.0310 | ns | ns |
| Filifactor | ns | 0.0343 | ns | ns | 0.0286 | ns | ns | 0.0316 | ns | ns |
| Atopobium | ns | 0.0004 | ns | ns | 0.0034 | ns | ns | ns | 0.0019 | ns |
| Alloprevotella | ns | 0.0005 | ns | ns | 0.0027 | ns | ns | 0.0050 | ns | ns |
| Pseudoramibacter | ns | 0.0231 | ns | ns | 0.0273 | ns | ns | ns | ns | ns |
| Campylobacter | ns | ns | 0.0004 | ns | ns | 0.0338 | ns | ns | ns | ns |
| Olsenella | ns | 0.0208 | ns | ns | ns | ns | ns | ns | ns | ns |
| Oribacterium | ns | <0.0001 | ns | ns | <0.0001 | ns | ns | 0.0089 | ns | ns |

Abundances at different sampling sites (U, upper extremities; L, lower extremities; H, head/neck; A, anogenital region; T, thorax/abdomen) were compared using the Kruskal–Wallis test with Dunn's multiple comparison post hoc test ($n = 104$). Given are genera, which significantly differed in abundance at least one sampling sites ($p < 0.05$). $p$-values > 0.05 are indicated by ns (non significant). In most cases, genera are of a significantly higher abundance at the later mentioned body site. Cases where genera are of a lower abundance at the later mentioned body site are indicated with bold numbers

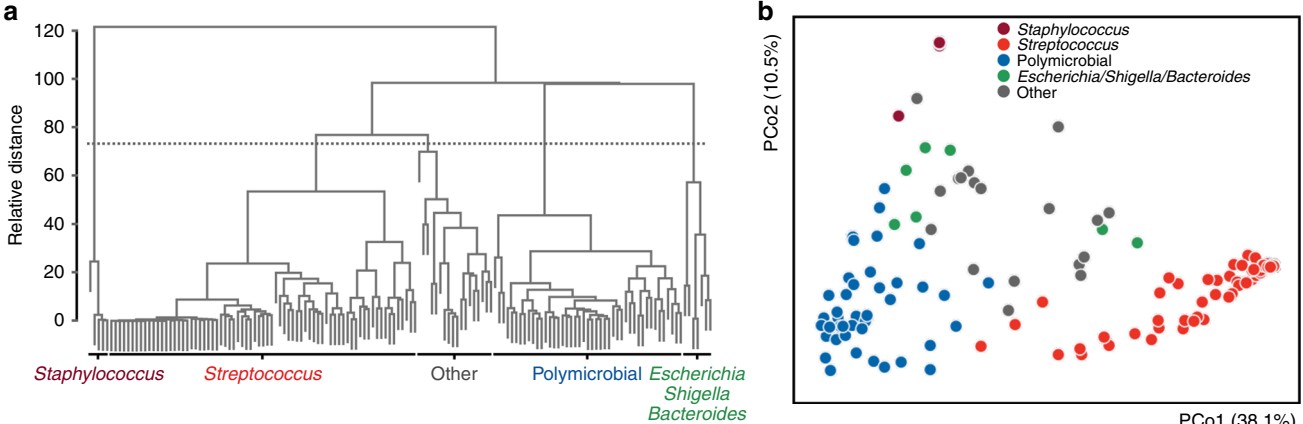

**Fig. 2** NSTIs can be grouped into distinct infection types based on the associated pathobiome. **a** Patient classification ($n = 148$) into different infection types based on their associated bacterial composition using hierarchical agglomerative clustering. Infection type definitions are assigned based on the genus-level distribution of the associated bacterial composition. **b** Principal Coordinate Analysis (PCoA) of Bray–Curtis dissimilarities between the bacterial composition identified in all tissue biopsies ($n = 148$), colored by infection type. Source data are provided in Supplementary Data 2 and 3

Due to their generally commensal lifestyle, the pathogenic mechanisms expressed by the bacterial species identified in polymicrobial NSTIs are severely understudied. To overcome this limitation, we assessed the pathogenic potential of pathobionts in polymicrobial NSTIs by querying the bacterial expression profiles for conserved virulence-associated protein domains, using the InterProScan tool (see Materials and Methods, Supplementary Data 6). To enable comparison between mono- and polymicrobial

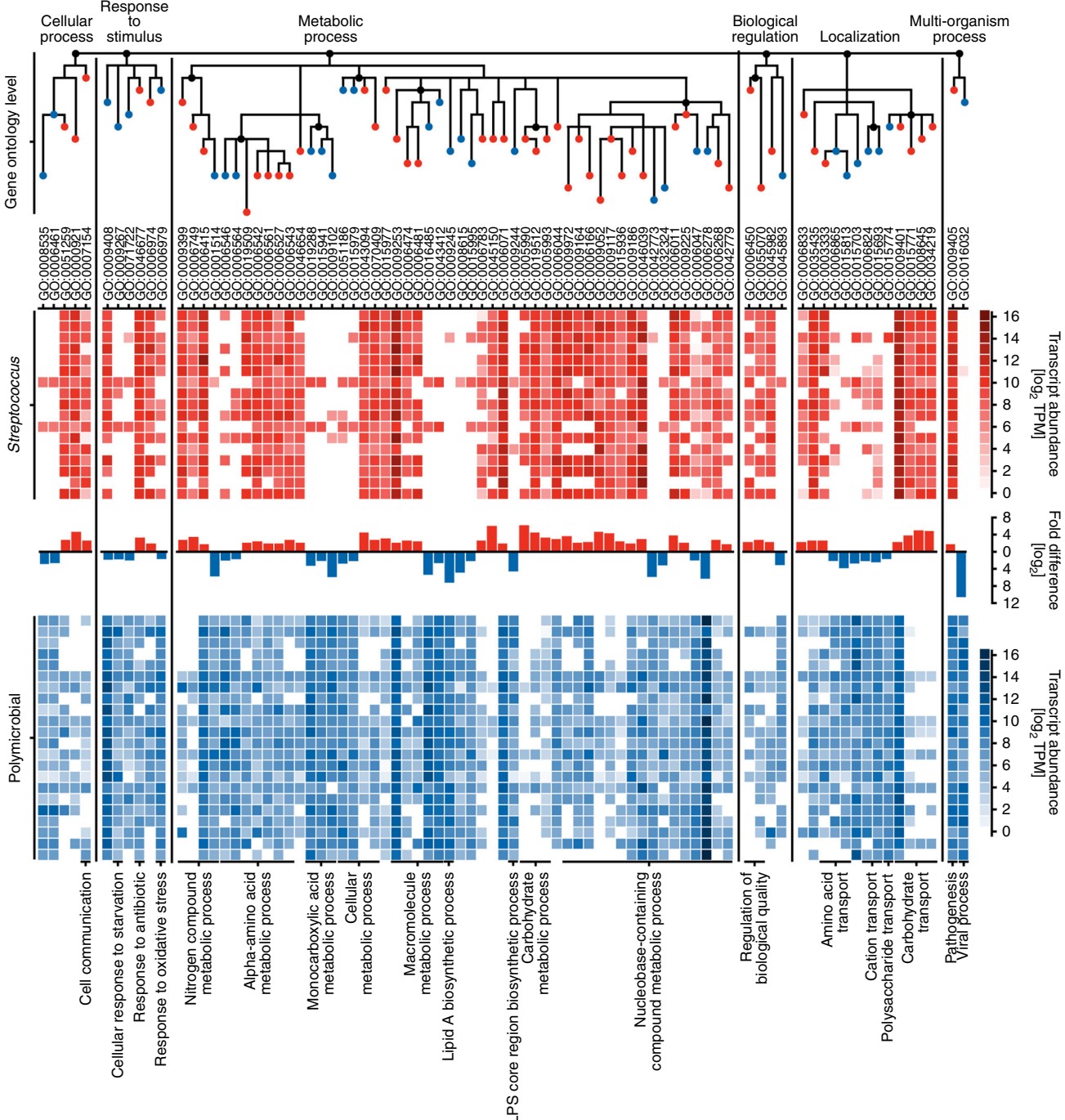

**Fig. 3** *Streptococcus* spp. and polymicrobial communities express different functionalities that facilitate nutrient acquisition and sustained inflammation. GO terms (root: biological process) with significantly higher associated gene expression in the transcriptional profile of *Streptococcus* spp. (red, $n = 17$) or polymicrobial communities (blue, $n = 22$) during NSTIs (Benjamini–Hochberg adjusted $p$-value $\leq 0.05$ Kruskal–Wallis test, $\log_2$ fold difference $\geq 2$) are shown. The heat map depicts summed sample-wise transcript abundance ($\log_2$ TPM) of genes associated with each GO term. The dendrogram depicts the GO hierarchy of all visualized GO terms. GO term clusters (labeled below the graph) are based on common parent terms. Nodes are hierarchically arranged to reflect distance to the root and their color indicates the pathobiome with the higher associated average expression. Source data are provided in Supplementary Data 5

NSTIs, the same method was applied to characterized virulence factors expressed by *Streptococcus* spp. in monomicrobial NSTIs.

Using this approach, we observed that genes encoding prominent virulence factors with well-defined roles in the pathogenesis of *Streptococcus* spp. during NSTI were highly expressed during NSTI (Fig. 4a, b). In contrast, expression of a greater diversity of virulence-associated domains was evident in polymicrobial NSTIs, reflecting the underlying diversity of the

bacterial community (Fig. 4a). Functional aggregation of the identified protein domains indicated that factors mediating adhesion/invasion, immune evasion, proteolysis, and toxin activities contribute to the pathogenicity of *Streptococcus* spp. in monomicrobial NSTIs, while only domains associated with factors mediating cellular adhesion and extracellular proteolytic activity were highly expressed by the bacterial communities associated with polymicrobial NSTIs (Fig. 4c, d).

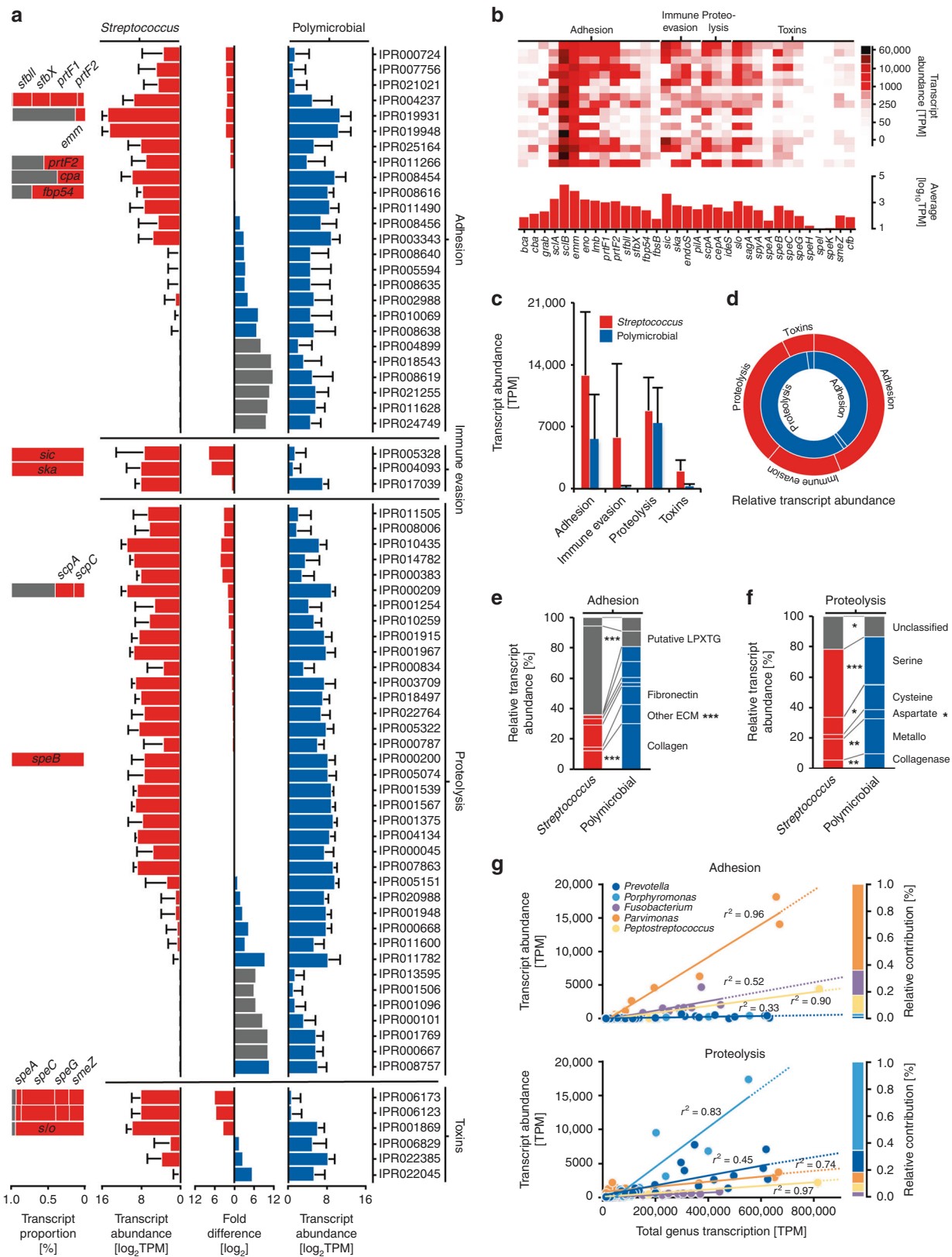

Interestingly, cellular adhesion, which facilitates tissue colonization and successful establishment of infection, was mediated by different mechanisms in monomicrobial *Streptococcus* spp. and polymicrobial NSTIs. For example, factors mediating adhesion to collagen and other extracellular matrix (ECM) components, known to be relevant for colonization of various host environments including commensal niches[20], were highly expressed by the polymicrobial pathobiome (Fig. 4e). On the other hand, expression of genes encoding fibronectin-binding proteins (*sfbII, prtF1, prtF2, fbp54*), which mediate not only adherence to ECM but also bacterial invasion into host cells[21], was pronounced in the transcriptional profile of *Streptococcus*

**Fig. 4** Different virulence functionalities contribute to the pathogenicity of *Streptococcus* spp. and polymicrobial communities during NSTIs. **a** Summed mean expression level ($\log_2$ TPM ± s.d.) of genes categorized based on their encoded InterPro (IPR) domains. The relative contribution of documented streptococcal virulence genes to IPR domain expression is depicted on the bars (left). The $\log_2$ differences in IPR expression between *Streptococcus* spp. (red, $n = 17$) and polymicrobial community (blue, $n = 22$) mediated NSTIs are depicted in the center. Gray bars mark IPR domains expressed by only one pathobiome and were calculated assuming one pseudo-TPM. **b** Streptococcal virulence factor expression (TPM) during NSTIs. The expression level is given for virulence-associated genes annotated using the virulence factor database (VFDB). **c, d** IPR domain expression associated with specific virulence categories. Average total abundances (TPM ± s.d.) (**c**) and relative expression levels of virulence categories (**d**) are given. **e, f** Relative proportion of InterPro domain expression in the functional categories 'adhesion' (**e**) and 'proteolysis' (**f**) associated with terms indicating target specificity (**e**) or protease class (**f**). Putative adhesins and unclassified proteases are depicted in gray. ***$p < 0.001$; **$p < 0.01$; *$p < 0.05$, Kruskal–Wallis test with Dunn's multiple comparison post hoc test ($n = 39$). **g** Contribution of the five genera with the highest average abundances in polymicrobial NSTIs to the total pathobiome expression of specific virulence categories. The relative amount of transcripts (TPM) expressed by each genus is plotted against the relative amount of transcripts encoding IPR domains involved in adhesion (top) or proteolysis (bottom). Squared correlation coefficients are depicted. Source data are available as a source data file

spp. (Fig. 4a, b, e). Similarly, while genes encoding a broad array of proteolytic factors were strongly expressed in both types of NSTIs, differences in the relative abundance of the protease families indicate that different proteolytic activities are exploited by *Streptococcus* spp. in monomicrobial NSTIs and the human pathobionts in polymicrobial NSTIs (Fig. 4f).

Importantly, during polymicrobial NSTIs different members of the polymicrobial community contributed to the expression of specific virulence functionalities to a different extent (Fig. 4g). Thus, *Porphyromonas* spp., which has been shown to express potent proteases in the oral cavity[22], contributed strongly to the proteolytic potential of the community, while *Parvimonas* spp., *Fusobacterium* spp., and *Peptostreptococcus* spp. were responsible for the majority of the expression mediating the adhesive properties of the pathobiome. These observations indicate that bacterial synergism may contribute to the pathogenicity in polymicrobial NSTIs.

**Host response differs in streptococcal and polymicrobial NSTIs.** The transcriptional analysis of infected tissue indicated that the gene expression profile differed significantly between monomicrobial *Streptococcus* spp. ($n = 17$) and polymicrobial ($n = 22$) NSTIs (PERMANOVA, $p$-value = 0.001), (Supplementary Figure 11, Supplementary Table 2). Nevertheless, a core acute inflammatory signature comprising genes encoding important inflammatory mediators, including cytokines (*IL-1β, IL-6, IL-8*), complement components (*C1q, C2, C3, C5*), alarmins (*S100A8/9*), genes encoding a range of factors with broad proteolytic activities (*ADAM8, 10, 12, 15, 17, 19, 28, ADAMTS2, 4, 5, 6, 9, 12, MMP2, 3, 8, 10, 12, 13, 15, 24, 25*) and hypoxia responsive genes (*HIF1A, ENO1, BHLHB2, BINP3, PHHA1, LDHA, GAPDH*) was observed in both mono- and polymicrobial NSTIs (Supplementary Data 7, 8). Among the genes with significantly greater expression in polymicrobial compared to monomicrobial NSTIs were those encoding ECM components like collagen, fibronectin or lumican, as well as connective tissue growth factor (*CTGF*) (Fig. 5a, b, Supplementary Data 8), proteins which are commonly expressed by activated fibroblasts[23]. On the other hand, a set of genes encoding interferon-inducible mediators such as *CXCL9, CXCL10, CXCL11, MX1*, and *MX2* as well as the guanylate-binding *GTP1* and *GTP2* were most prominently higher expressed in monomicrobial NSTIs, in particular those caused by *S. pyogenes* (Fig. 5a, Supplementary Data 8).

**Potential biomarkers for the diagnosis of *S. pyogenes* NSTIs.** To determine if the differences in interferon-inducible gene expression observed between streptococcal and polymicrobial NSTIs biopsies were also detectable at the protein level, a panel of interferon-related mediators including IFN-alpha, IFN-beta, IFN-gamma, IFN-lambda, IL-6, IL-8, IL-1beta, IL-10, TNF-alpha, GM-CSF, IL-28, CXCL9, CXCL10, and CXCL11 was measured in plasma samples of a set of NSTIs patients (training cohort) at the time of

hospital admission. This training cohort consisted of 12 monomicrobial *S. pyogenes* NSTI patients, 22 polymicrobial NSTI patients as well as five healthy control subjects (Supplementary Data 3). Of these proteins only CXCL9, CXCL10, and CXCL11 displayed statistically significant concentration differences between *S. pyogenes* and polymicrobial NSTIs (Fig. 6, Supplementary Figure 12).

Prompt diagnosis of group A streptococcal NSTIs is of outmost clinical importance, as these infections are frequently associated with high systemic toxicity, requiring rapid intervention to halt disease progression. Therefore, we assessed the potential use of host-derived signatures for microbial diagnosis of NSTIs caused by *S. pyogenes*. We compared the ability of three different modeling techniques (logistic regression, linear support vector machine and random forest classifier) to discriminate between *S. pyogenes* and non-*S. pyogenes* NSTIs based on the plasma concentrations of the measured protein-panel. The random forest classifier outperformed the other two approaches in the training cohort, achieving a mean area under the receiver operating characteristic curve (ROC-AUC) of 0.954 (Fig. 7a) and was therefore adopted for further analysis. Because in clinical practice it is desirable to keep the number of parameters required for diagnosis to a minimum and a reduction of feature space can also reduce model overfitting and improve performance, we applied the BORUTA algorithm to identify the most relevant features in the classification model. The algorithm identified the plasma concentrations of CXCL9, CXCL10 and CXCL11 as relevant for identification of the different types of NSTIs (Fig. 7b). The discriminatory power of a model trained only on these three parameters (3-feature model) was compared with that of the all-features model. The 3-feature model achieved a mean ROC curve AUC of 0.95 showing comparable performance to the full model (Fig. 7c). The potential of these plasma biomarkers for the identification of *S. pyogenes* NSTIs was then tested in an independent set of patients ($n = 59$), of which 27 were identified by 16S rRNA gene sequencing as monomicrobial *S. pyogenes* NSTIs and 32 as NSTIs caused by microorganism(s) other than *S. pyogenes* (Supplementary Data 3). We confirmed significantly higher plasma concentrations of CXCL9, CXCL10, and CXCL11 in *S. pyogenes* NSTI patients compared to NSTIs cases of other microbial etiology (Supplementary Figure 13). When tested in this independent cohort, the trained 3-feature random forest model achieved a ROC-AUC of 0.822 (Fig. 7d) and was able to correctly classify 21 of 32 non-*S. pyogenes* and 23 of 27 *S. pyogenes* NSTIs patients (Fig. 7e). Thus, while the model had a relatively high error rate in classifying non-streptococcal NSTIs, it exhibited promising performance in identifying *S. pyogenes* NSTIs (True Positive Rate 85%), highlighting the potential of a limited panel of host-derived biomarkers for rapid identification of *S. pyogenes* NSTIs allowing early therapeutical intervention.

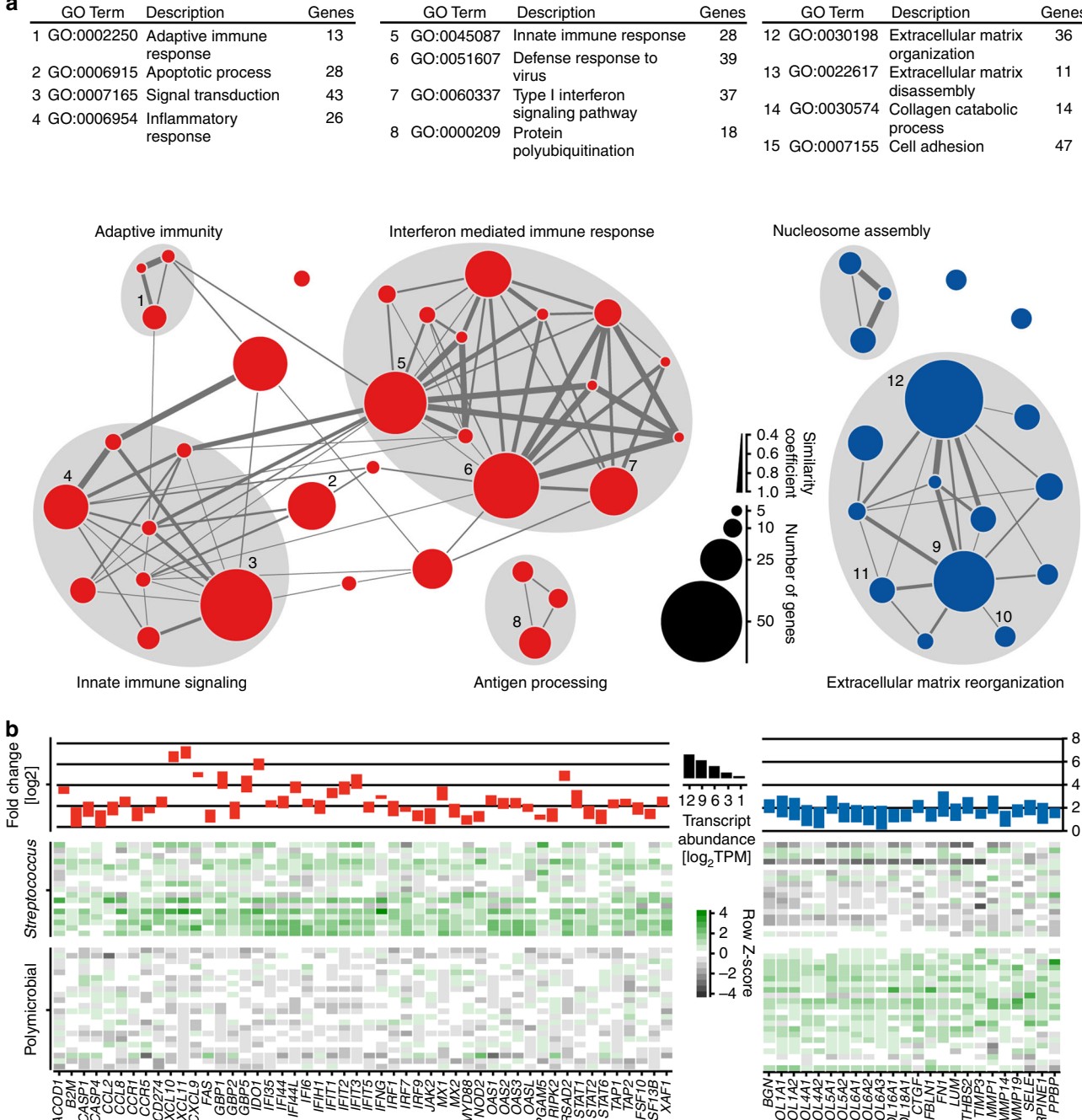

**Fig. 5** Functional profiling of infected patient tissue during streptococcal and polymicrobial NSTIs reveals distinct patterns of gene expression. **a** Network plot of gene ontology (GO) terms enriched within the genes differentially expressed between human tissue biopsies infected with *Streptococcus* spp. (red, $n = 17$) or the polymicrobial community (blue, $n = 22$) (Benjamini–Hochberg adjusted $p$-value ≤ 0.05, Wald test). Nodes represent enriched GO terms, grouped into functional clusters. **b** Average expression level, fold change (upper panel) and inter-individual transcriptional variation (lower panel) of selected genes with significantly higher transcript abundances in human tissue infected by *Streptococcus* spp. (red) or by a polymicrobial community (blue). Source data are provided in Supplementary Data 5, 7, and 8

## Discussion

It is well documented that NSTIs, besides being monomicrobial in nature and caused by professional pathogens, can also be caused by polymicrobial bacterial communities[1,7]. However, despite the fact that these polymicrobial infections often outnumber those of monomicrobial etiology, there is a lack of information regarding their bacterial composition and activities. Culture-dependent analysis of polymicrobial NSTIs remains generally restricted to the most abundant or easily culturable bacteria[8]. Amazingly, to date, only a single study has attempted to analyze the microbial composition in NSTIs using a 16S rRNA gene sequencing approach. This study, performed in a limited cohort of 10 volunteers, showed that standard cultivation methods underestimate the diversity of the complex bacterial communities in polymicrobial NSTIs[24]. In the study presented here, we used 16S rRNA gene sequencing to analyze the bacterial communities

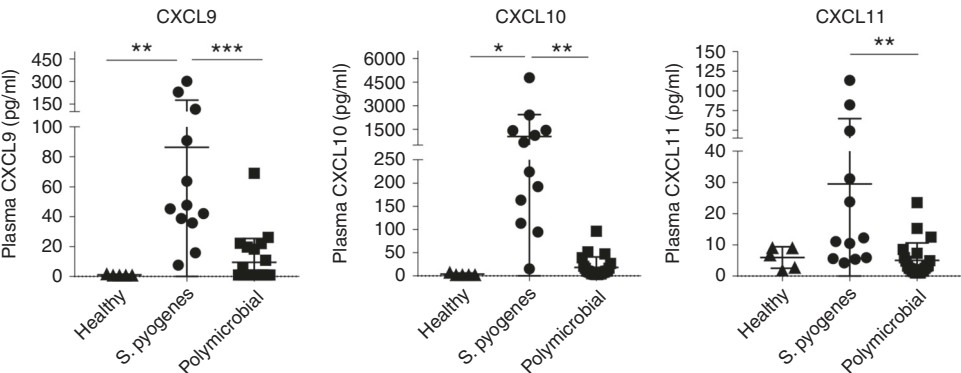

**Fig. 6** Plasma levels of interferon-inducible mediators that differ in NSTI patients. The levels of 15 interferon-inducible mediators were measured in the plasma of patients with *S. pyogenes* ($n = 12$) or polymicrobial ($n = 22$) NSTIs, or healthy controls ($n = 5$) by a multiplex beads array. The levels of those three mediators that differ between patients with polymicrobial and streptococcal NSTIs are shown whereas those of the 12 other mediators are given in Supplementary Figure 12. The mean value (±s.d.) is indicated by a horizontal line. Statistical significance was evaluated using ordinary one-way ANOVA with *$p$-value < 0.05; **$p$-value < 0.01; ***$p$-value < 0.001. Source data are available as a source data file

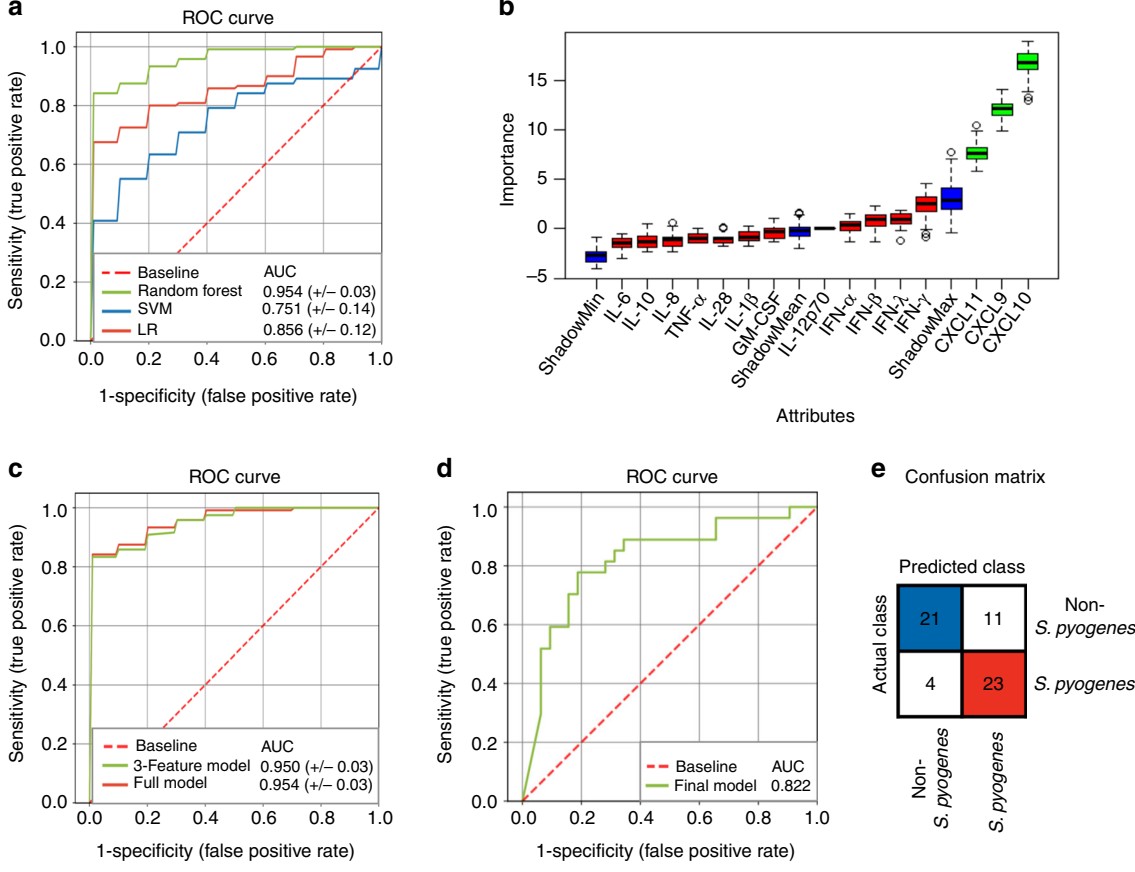

**Fig. 7** Identification of potential plasma biomarkers for microbial diagnosis of NSTIs. **a** ROC curves for the model comparison of Random Forest (RF, green), linear support vector machine (SVM, blue) and logistic regression (LR, red) on the training cohort ($n = 12$ *S. pyogenes* NSTIs, $n = 22$ non-*S. pyogenes* NSTIs) using the full panel of available measured variables. AUC values ± 95% CI are given. **b** Selection of relevant plasma markers for discrimination between *S. pyogenes* and non-*S. pyogenes* NSTIs in the training cohort using the Boruta algorithm. Boxplots of features are sorted by increasing importance according to the Z-scores. Features colored in green are those which were classified as relevant (exhibiting Z-scores higher than shadowMax). Features colored in red are unimportant for model performance. The blue boxes correspond to minimal (shadowMin), mean (shadowMean) and maximal (shadowMax) importance calculated from randomly permuted features. **c** ROC curves for a RF classifier trained on the full panel of features (red) and a 3-feature model trained solely on CXCL9, CXCL 10, and CXCL 11 (green) of the training dataset. AUC values ± 95% CI are given. **d** ROC curve showing the 3-feature RF classifier performance in the independent validation cohort ($n = 27$ *S. pyogenes* NSTIs, $n = 32$ non-*S. pyogenes* NSTIs). **e** Confusion matrix summarizing the performance of the 3-feature model in the independent validation cohort. Each row of the confusion matrix shows the number of samples in an actual class while each column shows the number of samples in a predicted class. Tiles showing the number of correctly classified cases are colored blue (non-*S. pyogenes*) or red (*S. pyogenes*). Source data are available as a source data file

associated with NSTIs in a cohort of 148 patients and could demonstrate that polymicrobial communities in NSTIs are not randomly associated, but typically comprised of co-occurring bacteria forming a highly interconnected network. The bacterial genera dominating in polymicrobial NSTIs, which include *Porphyromonas* spp., *Fusobacterium nucleatum* and *Prevotella* spp. (Fig. 1a, b, Supplementary Figure 8a) have been shown to synergistically interact in an in vitro model of chronic periodontitis[25] as well as in suppurative apical periodontitis in a murine infection model[26]. Hence, their co-occurrence during NSTIs might suggest that similar interactions facilitate tissue colonization and drive disease progression. Overall, these analyses refute the statement of Rudkjøbing et al. that there are no specific combinations of species in polymicrobial NSTIs[24]. Although several of the species that we identified in polymicrobial NSTIs, such as *Prevotella intermedia* or *Parvimonas micra*, have previously been recovered from other bacterial infections[27,28], they are also common members of the healthy human microbiota[29]. This observation indicates that the contribution of these microorganisms to the overall NSTIs burden is most probably still underestimated in the clinical setting as culture-based methods frequently fail in their detection[24,30]. Furthermore, the changing microbial epidemiology of NSTIs observed over the last decades[31] is also apparent in our our study. *Clostridium sensu stricto* were only observed in rare instances, a trend that has been linked to improvements in hygiene and sanitation conditions[32].

The results of the community profiling support the previously reported prominent role of *S. pyogenes* in the microbiology of NSTIs[7] as well as the high frequency of streptococcal NSTIs reported in Scandinavia[5]. The pathophysiology of *S. pyogenes* NSTIs has been extensively explored and many of the underlying pathogenic mechanisms previously described[10,33] are supported by our results. Thus, robust expression of adhesion factors such as fibronectin-binding proteins[34], of factors that enable the bacterium to circumvent the host immune response such as streptococcal inhibitor of complement (Sic), streptococcal C5a peptidase (ScpA)[35], and streptolysin O[36], as well as of important extracellular proteases such as SpeB[37] were observed. Adjunct treatment strategies to neutralize some of these factors such as IVIG therapy, which consist of pooled immunoglobulins from human donors have been suggested to limit destructive hyperinflammatory responses in NSTIs[2,38]. Although IVIG may be beneficial for treatment of *S. pyogenes* NSTIs[39], there is no evidence that it can provide therapeutic benefit for polymicrobial NSTIs[40]. Hence, there is a need for new therapeutic approaches that can be given as adjunct therapy for the treatment of polymicrobial NSTIs. In this study, we performed dual RNASeq analysis of human tissue biopsies to gain a more in-depth understanding of the microbial and molecular pathogenesis of NSTIs and to uncover targets for novel treatment strategies. While characterization of the metabolic activity and virulence of *S. pyogenes* in NSTIs was facilitated by the substantial existing knowledge about its pathogenicity mechanisms, the lack of a reference database of pathogenic factors for several bacteria identified in polymicrobial infections presented a significant limitation for this analysis. To overcome this limitation, we characterized the bacterial gene expression profile using conserved protein domains. We observed that, in contrast to *Streptococcus* spp. which rely on non-glucose carbohydrates metabolism as previously reported[41,42], the expression of genes encoding factors that contribute to amino acid transport and metabolism, as well as peptidolysis were strongly expressed by the polymicrobial communities. This suggests that nutrient acquisition by these bacteria might heavily rely on proteolytic degradation of host proteins as shown for some of the most abundant bacterial genera in polymicrobial NSTIs, *Fusobacterium* spp., *Prevotella* spp., and *Porphyromonas* spp., during periodontitis in the oral cavity[43]. Interestingly, we observed that *Porphyromonas* spp. contributed disproportionately to the proteolytic

potential of the microbial community. This observation supports the theory that synergistic complementation of specialized bacteria might be critical for establishment and progression of polymicrobial NSTIs. Additionally, a robust expression of genes coding for factors that contribute to lipopolysaccharide (LPS) synthesis was observed in polymicrobial NSTIs biopsies. This integral component of the outer membrane of Gram-negative bacteria is a powerful activator of inflammation in the human skin and the underlying tissues[44] and neutralization of LPS may provide an adjunct therapeutic strategy to decrease inflammation-induced pathology in polymicrobial NSTIs.

Because of the rapid disease progression, early diagnosis of NSTIs is critical to limit patient morbidity and mortality. Such a diagnosis, however, frequently proves to be challenging as symptoms can be ambiguous and frequently overlap with other clinical entities. Most prominently, the LRINEC score, comprising plasma concentrations of serum leukocytes, glucose, sodium, C-reactive protein (CRP), creatinine, and hemoglobin, has been developed to facilitate early diagnosis and limit time to surgical intervention[45]. Importantly, LRINEC scores did not differ between streptococcal and polymicrobial infections in our cohort and times from hospital admission to surgery were comparable ($53 \pm 114$ (mean $\pm$ s.d, $n = 73$) for streptococcal and $33 \pm 56$ ($n = 45$) hours for polymicrobial infections, Table 1). As recent work has questioned the diagnostic value of frequently used clinical parameters[46] and ultimate diagnosis can only be achieved by surgical exploration[1], diagnostic uncertainty of disease onset frequently hampers accurate early detection of NSTIs. This can lead to patients with variable stages of the disease being admitted to intensive care units, making prompt and accurate microbial diagnosis crucial to effectively halt disease progression. Guidelines for the management and treatment of NSTIs involve surgical debridement of the affected area combined with empiric high-dose intravenous broad-spectrum antibiotics to cover all potential pathogens that can cause NSTIs[11]. Antibiotic coverage can be narrowed once the causative microorganism(s) have been identified. For this reason, many NSTIs patients receive unnecessary or inappropriate antibiotics, which could be prevented by earlier identification of the causative pathogen(s). Therefore, the use of biomarkers that can distinguish between different types of NSTIs has the potential to reduce the time of broad-spectrum empirical therapy and to improve the patient outcomes by early administration of effective antibiotics. The comparative transcriptional profiling of NSTI tissue biopsies performed in this study uncovered interferon-related mediators, in particular type I interferon, associated with *S. pyogenes* NSTIs. This type I interferon response has been reported to be expressed by the host in response to *S. pyogenes* in order to reduce tissue damage caused by hyperinflammation[47]. We also observed significantly greater concentrations of the interferon-inducible chemokines CXCL9, CXCL10, and CXCL11 in the circulation of *S. pyogenes* NSTIs patients than in circulation of patients with polymicrobial NSTIs. The potential value of these chemokines as biomarkers of *S. pyogenes* NSTIs was shown by a random forest model trained on the plasma concentrations of these three chemokines followed by validation in an independent patient cohort. The results of this exploratory analysis identified plasma concentrations of CXCL9, CXCL10, and CXCL11 as being potentially useful biomarkers for microbial diagnosis of *S. pyogenes* NSTIs (85.2% true positive rate). While the classifiers showed a relatively high error rate when classifying non-*S. pyogenes* NSTIs (34.4%), early differentiation of NSTIs caused by *S. pyogenes* from those of different bacterial etiology is of outmost clinical importance as *S. pyogenes* NSTIs are generally associated with high systemic toxicity and require prompt intervention. Thus, interferon-induced chemokines show specific promise for the identification of patients requiring rapid ajunct therapies such as IVIG treatment[39]. The classifier performance may be improved in future studies that test a broader panel of plasma markers. Genes identified as differentially expressed between streptococcal and polymicrobial NSTIs in this

study might serve as a starting point to achieve higher classification accuracy via the inclusion of additional features in a classification model.

Following identification of a robust set of biomarkers, the challenge for the validation phase will be to develop suitable analytical methods capable for the fast quantification of the identified biomarkers. In this regard, microfluidic immunoassay systems in miniaturized devices that require small reagent volumes and very short analysis time may be a good option for generating such a NSTIs point-of-care diagnostic test. Several systems have been recently described combining microfluidics with detection of magnetic bead-based immunoassays for measurements in less than 1 h[48,49]. The multiplex microfluidic array designed by Malhotra et al.[48], for example, was able to detect four biomarker proteins in patient serum with high sensitivity and accuracy. We envisage a similar device for detection of CXCL9, CXCL10, CXCL11 or other host-derived biomarkers that could provide a rapid point-of-care serum test for microbial diagnostic and personalized therapy of NSTIs patients.

Taken together, our results demonstrate that despite similar clinical presentation, the pathophysiology of NSTIs caused by *S. pyogenes* or by polymicrobial communities differs significantly and that these differences can be eventually exploited for treatment and diagnostic purposes.

## Methods
**Study design**. This study was designed to identify the bacterial taxa associated with NSTIs, identify their functionalities that contribute to disease pathophysiology and identify host signatures of potential value for rapid microbial diagnosis at the time of hospital admission. Tissue biopsies and plasma samples were obtained from patients clinically diagnosed with NSTIs and enrolled in the EU-funded project INFECT (www.fp7infect.eu). Diagnosis of NSTIs was performed by the acting surgeon at primary operation or revision as described earlier[50]. Briefly, diagnosis was based on the presence of necrotic or deliquescent soft tissue with widespread undermining of the surrounding tissue. Following initial diagnosis, patient files and surgical descriptions were again scrutinized and patients were excluded from the study if reports of necrotic or deliquescent tissue were absent. All remaining patients were eligble to be included in this study. Written informed consent was obtained from all patients or their surrogate. The INFECT study is registered at ClinicalTrials.gov (NCT01790698). The INFECT study was conducted in accordance with the Declaration of Helsinki and with the approval of the regional Ethical Review Board at the Karolinska Institutet in Stockholm, Sweden (Ethics permits: 2012/2110-31/2), the National Committee on Health Research Ethics in Copenhagen, Denmark (Ethics permits: 1151739), the regional Ethical Review Board in Gothenburg, Sweden (Ethics permits: 930-12) and Bergen, Norway (2012/2227/REC West). Specimens from patients admitted to five Scandinavian hospitals, Righospitalet (Copenhagen, Denmark), Karolinska University Hospital (Stockholm, Sweden), Blekingesjukhuset (Karlskrona, Sweden), Sahlgrenska University Hospital (Gothenburg, Sweden), and Haukeland University Hospital (Bergen, Norway), were included in this study. All experiments were performed in full accordance with the approved ethics applications specified above. Patient mortality was recorded for a period of 365 days following study enrollment. The level of hemoglobin, white blood cells, C-reactive protein, creatinine sodium, and glucose in blood, patient pulse and mean arterial pressure were determined at hospital admission. Time from hospital admission to surgery was recorded for all patients. Only tissue samples taken on the day of hospital admission (day 0) were included in this study to enable the identification of robust host signatures indicative of bacterial etiology irrespective of time since onset of initial symptoms. Specimens for 16S rRNA gene sequencing (148) were randomly selected from patients with available day 0 tissue biopsies. All tissue biopsies were obtained intraoperatively and stabilized with RNAlater™ (Thermo Fisher Scientific).

**Sample pre-processing**. Stabilized tissue was transferred into 1 ml TRIzol™ (Invitrogen) reagent containing 50 μl of vanadyl ribonucleoside complex (VRC) and mechanically disrupted using a Polytron disperser (Kinematica). To achieve optimal bacterial lysis, samples were transferred into lysing matrices B (MP Biomedicals) and disrupted using a FastPrep-24® (MP Biomedicals). Tissue debris was removed by centrifugation and phase separation was achieved by addition of 200 μl Chloroform (Carl Roth) followed by centrifugation. The aqueous phase containing total RNA was separated for subsequent RNA purification, while the interphase and organic phase were used for further DNA purification.

**16S rRNA gene preparation and sequencing**. The DNA was purified following the manufacturer's protocol for TRIzol™ DNA extraction (Thermo Fisher

Scientific). Amplicon libraries of the V1–V2 region of the 16S rRNA gene were generated as previously described[51]. The target region was amplified by PCR following 20 cycles of PCR reaction using the 27F and 338R primers and sequenced on a MiSeq platform ($2 \times 250$ bp)[51].

**Bioinformatic analysis of the NSTI-associated microorganisms**. A total of 4,593,685 reads were generated, resulting in $31038 \pm 18665$ reads per sample. Bioinformatic processing was performed as previously described[51]. Raw reads were merged with the Ribosomal Database Project (RDP) assembler. Sequences were aligned within MOTHUR (gotoh algorithm using the SILVA reference database) and subjected to preclustering (diffs = 2) yielding phylotypes that were filtered for an average abundance of ≥0.001% and a sequence length ≥250 bp before analysis (Supplementary Data 1). Phylotypes were assigned to a taxonomic affiliation based on naïve Bayesian classification with a pseudo-bootstrap threshold of 80%. Phylotypes were then manually analyzed against the RDP database using the Seqmatch function to define the discriminatory power of each sequence type. Species level annotations were assigned to a phylotype when only 16S rRNA gene fragments of previously described isolates of a single species were aligned with a maximum of two mismatches with a representative sequence read. Similarly, genus-level annotations were assigned to a phylotype when only the representative sequence aligned to isolates and uncultured representatives of a single genus with up to two mismatches.

For additional taxonomic characterization, the identified *Streptococcus* spp., *Prevotella* spp., *Porphyromonas* spp., *Bacteroides* spp., *Fusobacterium* spp., and *Peptostreptococcus* spp. sequences were compared to sequences of all isolates with species level annotation available in RDP (release 11 update 5). Isolate sequences were aligned with MUSCLE[52] and edited using SEAVIEW[53]. Sequences were trimmed to the V1–V2 region and all sequences of poor quality (≥1 N per sequence) as well as those not completely covering the amplified region were deleted.

To reduce complexity, unique *Streptococcus* sequences assigned to only a single isolate with species level annotation were removed before phylogenetic analysis. Phylogenetic analyses were performed using the neighbor joining routine with Jukes–Cantor correction and pairwise deletion of gaps in MEGA7[54]. Species names were assigned to phylotypes that clustered with sequences of only a single species, were supported by high bootstrap values >50% and exhibited >97.5% of sequence identity to a reported sequence of a species representative. Phylotype data were rarified to resemble the smallest library size using the R phyloseq package (Supplementary Data 2) and used to construct sample-similarity matrices using the Bray–Curtis algorithm (V.7.0.11, PRIMER-E, Plymouth Marine Laboratory, UK). Bacterial alpha-diversity was calculated using the Gini–Simpson index (1-Λ) on rarified phylotype data.

NSTIs were classified according to the body part affected on admission into five categories: Head/neck, upper extremities, lower extremities, anogenital, and thorax/abdomen. Cases that had multiple body parts affected were excluded from the analysis. Significant differences between these five a priori defined groups of samples were evaluated using the permutational multivariate analysis of variance (PERMANOVA), allowing for type III (partial) sum of squares, fixed effects sum to zero for mixed terms and the generation of Monte Carlo p-values using unrestricted permutation of raw data. Phylotype and genera abundance as well as diversity indices across the five body parts were compared using the Kruskal–Wallis test with Dunn's multiple comparison post hoc test and were assumed to be significantly different if p-value ≤ 0.05.

**Bacterial co-occurrence modeling and specimen classification**. To identify patterns of bacterial co-occurrence an ensemble method implemented within the Cytoscape application CoNet, optimized for sparse datasets, was applied to column-wise normalized phylotype counts aggregated at the genus level[17]. To avoid misinterpretation resulting from sparse data, genera present in less than 25% of all samples were excluded prior to the modeling step. Two measurements of correlation between genera (Spearmen and Pearson) and two dissimilarity measurements (Kullback–Leibler and Bray–Curtis) were used to compute scores for genera interactions (network edges) ignoring parent-descendant relationships. Statistical significance was assigned to the resulting edge scores following measurement-specific permutation and bootstrapping with 999 iterations each. To counter the compositionality bias of correlation measures inferred from column-wise normalized data, a re-normalization approach that provides a null-distribution and captures the similarity introduced by compositionality was applied during the permutation step[17]. P-values of correlation measurements were calculated by z-scoring the permuted null and bootstrap confidence interval using pooled variance. In contrast, p-values of the dissimilarity measurements were computed by comparing the bootstrap confidence interval to a point null value computed by permutation, as they are inherently robust against compositionality[55]. The resulting measurement-specific networks were merged using Brown's method for p-value combination[56] and adjusted for multiple comparisons using the Benjamini–Hochberg correction. Phylotype interactions with adjusted p-values ≤ 0.05 and supported by at least two independent methods were retained in the final model (Supplementary Data 3). Distinct NSTI-associated bacterial communities were identified through divisive graph clustering of co-occuring genera using the Markov cluster algorithm (MCL)[57] with an inflation of 1.5 retaining modest

network granularity (Supplementary Figure 8a). The Markov clustering (MCL) algorithm was chosen as it has been successfully applied to a variety of biological datasets (microbiome data, gene networks, protein–protein interaction networks) and frequently outperforms other clustering algorithms[58]. Patients were classified into distinct types of NSTIs based on average-linkage hierarchical agglomerative clustering using the relative abundance of the identified bacterial communities (Fig. 2a). The optimal number of clusters in the resulting sample dendrogram was determined using the J-index[59] and distinct specimen clusters were defined to represent different types of NSTIs.

**RNA-seq sample preparation and sequencing**. Total RNA was extracted using the TRIzol™(Thermo Fisher Scientific) reagent according to the manufacturer's recommendations and DNA contamination was removed by DNase treatment using four U Turbo DNase™ (Ambion). RNA integrity was controlled on an Agilent 2100 Bioanalyzer (Agilent Technologies). ERCC RNA Spike-In Mixes (Thermo Fisher Scientific) were added to individual samples according to the manufacturer's guidelines for later determination of the lower limit of detection and to control for technical derived errors in the datasets. Ribosomal RNA was depleted using the RiboZero epidemiology kit (Illumina) according to the manufacturer's recommendations. cDNA libraries were prepared using the ScriptSeq v2 RNA-seq Library Preparation Kit (Illumina) according to the manufacturer's instructions. Libraries were size-selected between 250 and 1000 bp using BluePippin (Sage Science) following the manufacturer's protocol. Library quality and size-selection was controlled using an Agilent 2100 Bioanalyzer (Agilent Technologies). The cDNA libraries were sequenced ($1 \times 50$ bp) on the Illumina HiSeq 2500 platform to a maximum depth of 143.8 million reads using the TruSeq S.R. Cluster Kit, v3-cBot-HS (Illumina). Samples were exluded from further analysis if (1) the integrity of the isolated total RNA was low (RIN value < 4) or (2) the proportion of bacterial reads did not allow robust analysis of the generated dataset.

**Bioinformatic analysis of RNA-seq data**. Following demultiplexing and trimming, raw reads were pseudoaligned to the human genome assembly GRCh38.p10 using kallisto[60] and unaligned reads were stored for further bacterial metatranscriptome analysis. Human transcript based pseudoalignments were summarized at the gene level for later differential gene expression analysis using the R package tximport[61]. To eliminate potential confounding effects associated with variability in sample collection in a multi-center cohort study, unsupervised surrogated variable analysis was used to identify potential nuisance factors uncorrelated with the experimental factor of interest using the R package SVA[62]. Identified nuisance factors were included as covariates in downstream analysis. The R package DESeq2[63] was used to identify human genes that were differentially expressed between tissue biopsies from patients infected with *Streptococcus* spp. mono-infections and biopsies from patients with polymicrobial NSTIs. Significant nuisance factors identified by surrogate variable analysis were included as covariates in the design formula of the differential gene expression analysis. A gene was considered significantly differentially expressed if the Benjamini–Hochberg adjusted p-value ≤ 0.05. GO enrichment analysis was performed on all genes with significantly different transcript abundance using DAVID Informatics Resources[64]. Gene Ontology (GO) categories were considered significantly enriched in one infection type if the Benjamini–Hochberg adjusted p-value was ≤0.05. The functional relationship of significantly enriched GO terms was inferred from network analysis using EnrichmentMap[65]. Network edges were calculated using the overlap coefficient implemented in the package routine (threshold ≥0.4). Clusters of functionally related GO terms were identified by hierarchical clustering using the hierarchical clustering algorithm in protein interaction networks (HC-PIN, strong module definition, complex size threshold = 3) implemented in the CytoCluster package[66]. Absolute counts for each gene and each sample were then transformed into standard z-scores by subtracting the mean expression of a gene across all samples from the absolute counts in an individual sample and dividing this value by the standard deviation of the absolute counts of this gene across all samples.

Guided by the 16S rDNA amplicon analysis, genus-specific transcriptomes were compiled for all genera with a relative abundance ≥ 2% in at least one of the analyzed samples. Genus specific metatransriptomes were compiled from all available assemblies in the RefSeq database up to a maximum of $n = 250$ and including assemblies in a quality dependent hierarchical order: first complete genomes, thereafter chromosomes, scaffolds, and contigs (assessed August 2017). For each genus, assemblies were randomly selected if more than 250 genomes were available and lower quality assemblies (chromosomes, scaffolds, contigs) were only considered if the number of available genus genomes was ≤200 (Supplementary Data 9). Based on the genus composition of each specimen, sample-specific reference databases were created by concatenating the metatranscriptomes of all genera present with a relative abundance ≥2%. Reads that did not align to the human genome were pseudoaligned to their respective indexed reference using kallisto. Genes with low expression level (<11 counts in any sample) were excluded before downstream analysis. Redundant genes from different genomes, which were identified by matching NCBI RefSeq protein IDs, gene lengths and pseudoaligned read counts, were considered for further analysis. Genes were annotated (KEGG, GO, PFAM and InterPro) using InterProScan 5 (RC 1). Genus-specific functionality profiles were constructed by clustering translated gene sequences applying the UniRef50 parameters using BLAST (e-value ≤ $1 \times 10^{-6}$, sequence

identity ≥ 50%, coverage ≥ 80%), obtaining clusters of homologous genes[67]. Estimated counts, InterPro and NCBI annotations were aggregated for each gene cluster. The expression of each cluster of homologous genes was expressed as transcripts per million (TPM) for downstream analysis (Supplementary Data 10). Bacterial alpha-diversity for genus-level transcriptional contribution to functionalities (GO categories) was expressed using the Gini–Simpson index (1-Λ).

Differential expression of functional categories was determined by statistical comparison of the calculated log$_2$ fold change between bacterial communities using a Kruskal–Wallis test with Benjamini–Hochberg FDR correction for multiple comparisons. Functional categories were considered to be significantly differentially expressed if adjusted p-value ≤ 0.05 and log$_2$ fold change was ≥2. GO network graphs were constructed using the Python3 Networkx module, considering all nodes and connecting edges along the longest path to the biological process root node (GO:0008150). For analysis of the bacterial metabolic profile, redundant KEGG pathway annotations were removed by selecting parsimonious KEGG pathway sets based on the KEGG link database (release 85.1) using MinPath 1.4[68].

**Virulence factor gene level annotation**. The majority of highly abundant genera associated with polymicrobial infections were underrepresented or non-existing in relevant virulence factor databases, hindering analysis of the virulence mechanisms contributing to disease pathogenesis. To mitigate this problem, a strategy was developed to analyze the virulence potential of the bacterial communities in NSTIs based on conserved protein domains of gene products. A library of InterPro domains associated with the terms 'toxin', 'innate immune evasion', 'proteolysis' and 'adhesion' was generated (Supplementary Data 6), where InterPro domains were sorted into one of these categories, if; (I) a relevant GO term was associated with the domain (GO terms 'proteolysis' (GO:0006508), 'pathogenesis' (GO:0009405), 'adhesion' (GO:0022610), 'cell killing' (GO:0001906), and their respective child terms), (II) a gene product associated with a respective domain had experimentally validated mechanistic relevance for one of the categories, (III) the InterPro domain description showed semantic association to one of these disease relevant processes (e.g. immune evasion). The InterPro domain architecture of all expressed gene clusters was queried for InterPro domains that were classified into one of the analyzed virulence-associated categories. The relevance of all identified features for host-pathogen interaction was further inferred by in silico prediction of their cellular export using SignalP 4.0, which identifies signal peptides for cellular export using neural networks (D-cutoff values 'sensitive')[69], and SecretomeP 2.0, predicting non-signal peptide mediated protein secretion ab initio[70]. The validity of the above-mentioned strategy for identifying potential virulence features in bacterial communities was demonstrated using well-known virulence and virulence-associated gene clusters of S. pyogenes annotated by querying the Virulence Factor Database VFDB (December 2017)[70]. Virulence factors from S. pyogenes, S. dysgalactiae and S. agalactiae and their cognate locus tags were extracted from the VFDB and used to identify respective gene clusters.

**Quantitative RT-PCR**. The expression of a set of genes indicated by RNA-Seq as differentially expressed between monomicrobial and polymicrobial NSTIs was further evaluated by qRT-PCR (Supplementary Figure 14), Total RNA samples were reverse transcribed and amplified using the SensiFAST™ SYBR® No-ROX Kit (Bioline) following the manufacturer's instructions. Primer sequences are provided in Supplementary Table 3. Thermal cycling conditions for all reactions were the following: reverse transcription for 20 min at 45 °C, initial denaturation for 5 min at 95 °C, 40 cycles of 20 s at 95 °C, 20 s at 60 °C and 20 s at 72 °C. Fold change values were calculated using Pfaffl's method for relative quantification in real-time RT-PCR[71].

**Determination of cytokines and chemokines concentrations**. Plasma concentrations of cytokines/chemokines were determined using the bead-based multiplex immunoassay Legendplex™ (Biolegend) according to the manufacturer's protocol. Available plasma samples taken from all patients included in the RNA-seq analysis on day 0 were used in the multiplex imunnoassay.

**Statistical analysis of plasma markers**. The Student t test and One-way ANOVA were used to evaluate the significance of differences in plasma marker concentrations between the different patient groups. Model performance was evaluated based on a training cohort, not considering samples with missing values, through $10 \times 2$ repeated stratified K-fold crossvalidation to ensure representation of the minority class in each sample split. Model parameters were optimized through a grid search approach to maximize the correct classification of the positive class (streptococcal NSTIs), represented by recall and precision score. Based on grid search L2-regularized Logistic Regression classifier (C: 1.0) and SVM classifier (kernel: linear, C: 10.0, gamma: 1.0) for standard scaled data, as well as Random Forest classifier (number of trees: 250, max features at each split: sqrt(n_features), minimal terminal leaf size: 3) for unscaled data were included in model comparisons. Feature selection was conducted using the Boruta algorithm which works as a wrapper algorithm around random forests (RF), performs a test for statistical significance of variable importances compared to a permuted copy of the original dataset (shadow dataset) and outputs a distribution of Z-scores for each feature[72].

Only the features with Z-scores statistically higher than the maximum achieved distribution for the shadow dataset were considered important. The final model was trained on the whole training cohort considering only the most important features (CXCL9, CXCL 10, CXCL11) and was evaluated using an independent test cohort of NSTI patients (32 polymicrobial non-streptococcal and 27 *S. pyogenes* NSTIs, see Supplementary Data 3) that had been kept separate during model training. For evaluation of the classifier performance, receiver operation characteristic (ROC) curves and the corresponding area under the curve (AUC) were built by plotting the true positive rate (sensitivity) against the false positive rate (1-specificity) for every likely cutoff between the 2 classes. Models were constructed, assessed and validated using the Python package 'sklearn'. Statistical analysis and plotting were done using the Python package matplotlib and Graph Pad Prism (Graph Pad Software, Inc.).

**Reporting summary**. Further information on research design is available in the Nature Research Reporting Summary linked to this article.

## Data availability

The sequencing data supporting the findings of this study have been deposited in the NCBI BioProject database with accession number PRJNA479582. The source data underlying all main text and supplementary figures are provided as a Source Data file. All other relevant data are available as Supplementary Data or available from the corresponding author.

## Code availability

The analysis underlying Figs. 3, 4b and 7 have been implemented in Python and R, and the source code can be downloaded from github [https://github.com/MolProfileStrepNSTI/NSTI_src_code].

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

## Acknowledgements

The authors thank A. Dröge S. Kahl, I. Plumeier and K. Mummenbrauer for technical assistance. They also thank E. Keen, O. Goldmann and members of the Microbial Interactions and Processes Research Group for discussions and comments on the manuscript. This research was supported by the European Union Seventh Framework Program (grant 305340, Infect) and by iMed, the Helmholtz Association Initiative on Personalized Medicine.

## Author contributions

D.H.P., R.T., A.I. and E. M. conceived and designed the study. R.T., A.I., E.M. and D.H.P. designed and supervised the experiments. R.T., E.M. D.H. and A.I. performed the experiments. R.T., A.I., E.M., J.H. and D.H.P. analyzed the data. M.B.M., O.H., S.S., A.N.T., T.B. and the INFECT Study Group performed patient recruitment and patient sample collection. R.T., E.M., A.I. and D.H.P. prepared and wrote the manuscript. All authors commented on the manuscript.

## Additional information

## INFECT study group

Oddvar Oppegaard[4], Eivind Rath[4], Torbjørn Nedrebø[4], Per Arnell[8], Anders Rosen[8], Peter Polzik[3], Marco Bo Hansen[3], Mattias Svensson[6], Johanna Snäll[6], Ylva Karlsson[9] & Michael Nekludov[10]

[8]Department of Anaesthesia and Intensive Care, Sahlgrenska University Hospital Ostra, Gothenburg, Sweden. [9]Department of Anaesthesia and Intensive Care, Blekingesjukhuset, Karlskrona, Sweden. [10]Department of Physiology and Pharmacology, Section for Anaesthesiology, Karolinska University Hospital, Stockholm, Sweden

