## [Peer Review File · Nature Communications]

Reviewers' comments:

Reviewer #1 (Remarks to the Author):

This manuscript studies Necrotizing soft tissue infections (NSTIs) by jointly performing microbial community profiling (via 16S rRNA gene sequencing) and transcriptional analysis (dual RNA-seq) in patient biopsies. The motivation behind this study is to better understand the pathophysiology of mono- and polymicrobial NSTIs with the ultimate goal of improving the early diagnosis/identification of the causative pathogens.

More specifically the authors first performed 16S rRNA gene sequencing in n=148 patient biopsies to identify their microbial composition and then selected n=17 monomicrobial and n=22 polymicrobial NSTIs samples for transcriptional profiling with dual RNA-seq. Based on their initial analysis, the authors then focused on n=12 out of the 17 monomicrobial NSTIs and aimed to train a binary classifier that is able to predict whether or not patient NSTIs are caused by *S. pyogenes* based on three protein markers in plasma samples.

Overall the concept of integrating microbial community identification with transcriptional profiling is very interesting. The dataset generated by the authors is informative and the corresponding bioinformatics analysis, albeit basic from a methods perspective -- is certainly able to provide high-level insights and guidelines for follow-up studies on polymicrobial NSTIs. However, it seems that the authors have only performed simultaneous transcriptional profiling on 39 out of 148 samples which significantly limits the scope of their paper in terms of reusability/validation. Presenting and analyzing a complete multimodal dataset consisting of matching 16S rRNA gene sequencing, dual RNA-seq and plasma panels from all 148 NSTIs patients would greatly increase the impact of this study, allowing for independent data analysis and method development. This would make the paper appealing to a much wider audience, as it is expected from NCOMMS articles, by serving as a comprehensive reference point for "integrating microbial community profiling with host and pathogen transcriptional analysis in patient biopsies".

From a bioinformatics analysis perspective the manuscript is relatively straightforward but technically sound as far as I can tell. The authors refer to existing methodology for their analyses and use well-established bioinformatics tools to generate their results. However a major concern is in regard to the approach of "identifying a set of diagnostic markers" and the use of the term "machine learning" to describe it (e.g., in the abstract). I would strongly advise the authors to tone down the machine learning aspect of their analysis as it currently may be misinterpreted by some readers as an attempt to impress. In fact, the approach considered by the authors never used "machine learning" to identify the markers; a set of candidate markers was identified based on differential expression statistics and as far as I can tell the three chemokines were chosen manually as the ones that were (individually) sufficiently different. After the three chemokines were chosen, it seems that the authors manually evaluated models that include only two out of the three chemokines (it is not clear if all 3 choose 2 combinations were tested since there is only one red curve in Fig7b) against a model that simultaneously contains all three, and concluded that the latter has a better classification performance based on the resulting ROC curves. The authors should consider a more principled approach to feature selection based on all measured variables (entire panel). Some common approaches for random forests in particular can be found for example in [Frauke Degenhardt, Stephan Seifert, Silke Szymczak; Evaluation of variable selection methods for random forests and omics data sets, Briefings in Bioinformatics, 2017]. Moreover the authors do not justify the use random forest classifiers to begin with, instead of say logistic regression or even kernel based svms. As a matter of fact regularized (e.g., L1 penalized) linear classifiers could provide a principled way of selecting variables and have been shown to generalize well in terms of test error. If the authors tested this and did not work they should include it in the results. A more comprehensive justification and evaluation of methods is definitely needed in my opinion before this paper can meet the standards of an NCOMMS article -- especially since the developed classifier is not performing particularly well.

Overall I think that the paper builds a very interesting case for better understanding NSTIs, and provides a basic yet informative analysis on the underlying pathophysiology. From an exploratory analysis perspective, this study is certainly able to achieve its goal. However it is the last part of the paper that I find to be its major weakness. Both the methodology and the limitations of the resulting classifier (e.g., only discriminating between *S. pyogenes* vs rest) fail to meet the expectations raised by the title/abstract. If the classifier performance cannot be improved then the authors should revise their statements to make this clear throughout the paper. In that case the authors should only emphasize the *exploratory* analysis part of their work, address its limitations and use the last part of the paper as an example illustrating its *potential* diagnostic value.

For more specific comments and questions please see below:

- 1- I think that the title should be more specific; If the "unique signatures for diagnosis" refers to the classifier then it should be made clear that this is only for detecting *S. pyogenes*. Similarly the abstract should be revised to better reflect the authors results/contributions.
- 2- The authors state in the introduction (and in the abstract) that "rapid and accurate identification [...] is critical for initiation of the appropriate antimicrobial therapy". To me this seems to be the main goal of the study. Although it may be straightforward for medical practitioners/researchers, the authors should include more details so that a reader outside the field can understand the potential impact of this line of work. In the best case scenario, how would the authors/medical practitioners use a "black box" that could immediately output a list of the causative organisms for a given infection? How would this change existing therapeutic approaches. How specific is current antibiotic therapy?
- 3- What is the state-of-the-art/current approaches in detecting mono and poly microbial NSTIs?
- 4- It seems that the authors' binary classifier is only able to detect whether an infection is caused by *S. pyogenes* or not. This is very far from the ultimate goal. How useful would this be in practice? Even if you could do this with perfect accuracy (a performance that the current classifier is still very far from achieving) it seems that the information gain is very limited. E.g., a-priori assuming that all NSTIs are polymicrobial will be correct in ~85% of the cases.
- 5- Please justify and clarify the definition of mono- and polymicrobial NSTIs (p.6 line 108). Is this the standard definition? I may be missing something here but what happens if Gini-Simpson diversity is between 0.25 and 0.5?
- 6- Please point to the methods in p7 line 131 to explain what divisive clustering of co-occurring genera means. How robust is this clustering to other clustering algorithms?
- 7- Please be more specific in the caption of Fig.1c (e.g., say what dots represent)
- 8- line 137, Fig 1c Please provide the corresponding statistic to justify significance.
- 9- Several figures only show the average values (mean). Is this representative of the underlying population? It would be more informative to show the standard deviation or confidence intervals with respect to the mean. This may be hard for some figures e.g., stacked barplots but it would certainly give a better understanding of the corresponding data.
- 10- Fig2b: PCoA -> Principal Coordinates Analysis (PCoA)

11- line 151: shouldn't this be just microbial community composition? If not then please provide more details.

12- line 154: "we randomly selected". Please point to the supplementary table and also state how many.

13- I understand why only 17 monomicrobial NSTIS were chosen but one of my main concerns was that only 22 polymicrobial NSTIs were analyzed. Please explain if there is a particular reason behind this choice.

14- line 156: Fig2. Is this the correct reference/figure?

15- line 163, line 108: This may be a naive question: Is there a reason why two measures of diversity are used in the paper? (alpha-diversity, Gini-Simpson diversity index).

16- line 168-178: Better annotation is needed. E.g., I couldn't find oligosaccharide core region biosynthesis, lipid A biosynthesis, polysaccharide transport or LPS annotated in Fig 3 or Supp Table 7.

17- line 268: How did you calculate 85% accuracy? Figure 7f shows 21 black dots that were correctly classified (predicted probabilities >0.5) out of 27. This gives $21/27 = \sim 77\%$ accuracy. Overall this entire section should be revised and more rigorous analysis/evaluation of the classifiers should be performed (see my initial comments).

Reviewer #2 (Remarks to the Author):

This is a timely and well designed study that will provide valuable information for clinicians and researchers. The methodology is State of the Art. These are of course devastating infections and the authors suggest a 70% mortality. This is probably much higher than other studies they cite. Still without proper diagnosis and emergent intervention, that may be true.

Few publications have had such a comprehensive evaluation of the microbes and potential pathogenic mechanisms. The authors are to be commended.

The authors have defined their studies according to a zero time and this is the point they obtained specimens from blood and operative procedures.

I can understand their definition. Yet numerous studies, which they cite, all suggest that the time to surgery is critical in determining outcome.

What was the mortality in this study?

What was the time from onset of signs and symptoms to operative procedures? They define this as zero time. I do not think that is sufficient.

What for the purpose of the study was their definition of a necrotizing infection? Was this based on histopathology? If so what were those findings. Surgeons impression, gram stains, evidence of septic shock or path reports? These should be included in the criteria for entrance into the study.

The include a reference to a study by Olsen and Musser that states that proteolytic activity in group A streptococcal necrotizing infection was important. Yet their results describe proteolytic activity as being more representative of poly-microbial necrotizing infections.

Those data that suggest that interferon mediators allow, differentiation between polymicrobial and group A streptococcal infection may be statistically significant, there is great overlap in quantitative results. So in a given patient this may not be diagnostically accurate, especially if one considers the time of onset of infection, and they only provide one data point which they define as zero time.

I consider these important issues that should be addressed.

Response to reviewer's comments:

Reviewer #1

(Remarks to the Author): This manuscript studies Necrotizing soft tissue infections (NSTIs) by jointly performing microbial community profiling (via 16S rRNA gene sequencing) and transcriptional analysis (dual RNA-seq) in patient biopsies. The motivation behind this study is to better understand the pathophysiology of mono- and polymicrobial NSTIs with the ultimate goal of improving the early diagnosis/identification of the causative pathogens. More specifically the authors first performed 16S rRNA gene sequencing in n=148 patient biopsies to identify their microbial composition and then selected n=17 monomicrobial and n=22 polymicrobial NSTIs samples for transcriptional profiling with dual RNA-seq. Based on their initial analysis, the authors then focused on n=12 out of the 17 monomicrobial NSTIs and aimed to train a binary classifier that is able to predict whether or not patient NSTIs are caused by S. pyogenes based on three protein markers in plasma samples. Overall the concept of integrating microbial community identification with transcriptional profiling is very interesting. The dataset generated by the authors is informative and the corresponding bioinformatics analysis, albeit basic from a methods perspective -- is certainly able to provide high-level insights and guidelines for follow-up studies on polymicrobial NSTIs. However, it seems that the authors have only performed simultaneous transcriptional profiling on 39 out of 148 samples which significantly limits the scope of their paper in terms of reusability/validation. Presenting and analyzing a complete multimodal dataset consisting of matching 16S rRNA gene sequencing, dual RNA-seq and plasma panels from all 148 NSTIs patients would greatly increase the impact of this study, allowing for independent data analysis and method development. This would make the paper appealing to a much wider audience, as it is expected from NCOMMS articles, by serving as a comprehensive reference point for "integrating microbial community profiling with host and pathogen transcriptional analysis in patient biopsies".

Answer: We agree with the reviewer that a complete multimodal dataset would have been optimal, however there are several reasons why we were unable to generate a more comprehensive dual RNAseq dataset. First it has to be mentioned that this study, to our knowledge, represents one of the first studies reporting *in vivo* dual RNAseq data of a bacterial infection in human patients. This is probably mainly due to the technical problems associated with obtaining high quality datasets for both human host and bacterial pathogen from the same tissue sample. Generally, bacterial mRNA abundance in the infected tissue is several orders of magnitude lower than human mRNA, which often results in totalRNA pools where the overwhelming majority (frequently >99%) is made up by eukaryotic RNA (see Westermann et al. PLoS Pathogens, 2017). When working with unique patient samples, this problem inherently limits the pool of samples that provide data of sufficient quality for in depth transcriptional analysis of both bacteria and eukaryotic host and requires cost-intensive deep sequencing of each sample (frequently one/two samples per HighSeq lane). These technical problems are amplified in the case of NSTIs where, in addition to the low bacterial/host mRNA ratio, tissue necrosis can hamper extraction of high-quality RNA and the generation of human transcriptional data of sufficient quality. During the technical optimization performed in this study, we frequently encountered this problem leading us to exclude several samples that exhibited insufficient quality of the generated datasets. A sentence has been included in the revised version of the manuscript addressing sample exclusion criteria (see Methods – ‘RNA-seq sample preparation and sequencing’). These technical limitations as well as the immense costs associated with *in vivo* dual RNAseq were obviously appreciated by both reviewers, and, as stated above, might be one reason why studies presenting such datasets from human cohorts remain rare. It is, however, important to point out that while we might underestimate transcriptional differences in the signature associated with different types of NSTIs because of reduced sensitivity, this does not invalidate the conclusions of the study. The insights gained from this

study represent a leap for understanding particularly the pathophysiology of polymicrobial NSTIs.

From a bioinformatics analysis perspective the manuscript is relatively straightforward but technically sound as far as I can tell. The authors refer to existing methodology for their analyses and use well-established bioinformatics tools to generate their results. However a major concern is in regard to the approach of "identifying a set of diagnostic markers" and the use of the term "machine learning" to describe it (e.g., in the abstract). I would strongly advise the authors to tone down the machine learning aspect of their analysis as it currently may be misinterpreted by some readers as an attempt to impress. In fact, the approach considered by the authors never used "machine learning" to identify the markers; a set of candidate markers was identified based on differential expression statistics and as far as I can tell the three chemokines were chosen manually as the ones that were (individually) sufficiently different. After the three chemokines were chosen, it seems that the authors manually evaluated models that include only two out of the three chemokines (it is not clear if all 3 choose 2 combinations were tested since there is only one red curve in Fig7b) against a model that simultaneously contains all three, and concluded that the latter has a better classification performance based on the resulting ROC curves. The authors should consider a more principled approach to feature selection based on all measured variables (entire panel). Some common approaches for random forests in particular can be found for example in [Frauke Degenhardt, Stephan Seifert, Silke Szymczak; Evaluation of variable selection methods for random forests and omics data sets, Briefings in Bioinformatics, 2017]. Moreover the authors do not justify the use random forest classifiers to begin with, instead of say logistic regression or even kernel based svms. As a matter of fact regularized (e.g., L1 penalized) linear classifiers could provide a principled way of selecting variables and have been shown to generalize well in terms of test error. If the authors tested this and did not work they should include it in the results. A more comprehensive justification and evaluation of methods is definitely needed in my opinion before this paper can meet the standards of an NCOMMS article -- especially since the developed classifier is not performing particularly well.

Answer: We truly appreciate the reviewer's helpful comments. We have toned down the machine learning aspect, included a comparative analysis of different modeling techniques (logistic regression, linear support vector machine and random forest - see Fig. 7a) to validate our use of random forest and applied a method for feature selection (Boruta) using the entire panel as recommended. We chose the Boruta algorithm because it captures all features in a dataset which are strongly or weakly relevant to the outcome variable (type of NSTI in our case). Boruta uses a wrapper approach built around a random forest classifier and progressively eliminates features in each interaction which did not perform well. A new panel in Fig. 7 (Fig. 7b) has been added to the revised manuscript showing the features sorted by increasing importance according to the Z-scores computed by the Boruta algorithm. We additionally compared performance of the sparse model, build solely on the features identified via feature selection, and the full model to justify our use of this sparse model for further analysis. The Method section has been edited to include more comprehensive information on our approach.

*Overall I think that the paper builds a very interesting case for better understanding NSTIs, and provides a basic yet informative analysis on the underlying pathophysiology. From an exploratory analysis perspective, this study is certainly able to achieve its goal. However it is the last part of the paper that I find to be its major weakness. Both the methodology and the limitations of the resulting classifier (e.g., only discriminating between *S. pyogenes* vs rest) fail to meet the expectations raised by the title/abstract. If the classifier performance cannot be improved then the authors should revise their statements to make this clear throughout*

the paper. In that case the authors should only emphasize the ***exploratory*** analysis part of their work, address its limitations and use the last part of the paper as an example illustrating its ***potential*** diagnostic value.

Answer: We agree with the reviewer regarding the "exploratory" nature of our study in regard to the identified biomarkers. We have highlighted this aspect in the Results and Discussion section and adjusted the title/abstract accordingly.

For more specific comments and questions please see below:

*1- I think that the title should be more specific; If the "unique signatures for diagnosis" refers to the classifier then it should be made clear that this is only for detecting *S. pyogenes*.*

Similarly the abstract should be revised to better reflect the authors results/contributions.

Answer: We have edited the title to more precisely point out the main conclusion of our study. We additionally modified the abstract to more specifically point towards our main conclusions/contributions to the field.

2- The authors state in the introduction (and in the abstract) that "rapid and accurate identification [...] is critical for initiation of the appropriate antimicrobial therapy". To me this seems to be the main goal of the study. Although it may be straightforward for medical practitioners/researchers, the authors should include more details so that a reader outside the field can understand the potential impact of this line of work. In the best case scenario, how would the authors/medical practitioners use a "black box" that could immediately output a list of the causative organisms for a given infection? How would this change existing therapeutic approaches. How specific is current antibiotic therapy?

Answer: We restructured the introduction and added a paragraph to give readers from outside the medical field more comprehensive background information on commonly used empiric treatment for NSTIs. We have also included a paragraph in the Introduction section indicating how the results of our study could contribute to improve microbial diagnostic of NSTIs with the concomitant increase in time-effectiveness of therapeutic intervention.

3- What is the state-of-the-art/current approaches in detecting mono and poly microbial NSTIs?

Answer: A concise description of the approach (tissue and blood cultures) currently used for microbial diagnosis of NSTIs has been included in the Introduction of the revised manuscript.

*4- It seems that the authors' binary classifier is only able to detect whether an infection is caused by *S. pyogenes* or not. This is very far from the ultimate goal. How useful would this be in practice? Even if you could do this with perfect accuracy (a performance that the current classifier is still very far from achieving) it seems that the information gain is very limited. E.g., a-priori assuming that all NSTIs are polymicrobial will be correct in ~85% of the cases.*

Answer: We agree with the reviewer that the identified signature is not sufficient to be used as the sole method of microbial diagnosis for NSTIs. However, one of the main problems with current methods used for microbial diagnosis is their long turnaround time. Given that NSTIs are rapidly progressing infections that can spread from a locally limited infection to a widespread necrosis requiring immediate amputation of affected body parts, reducing diagnostic turnaround time is a major imperative. This seems particularly important for

streptococcal NSTIs that are, in contrast to other types of NSTIs, associated with high levels of systemic toxicity (that can cause toxic shock syndrome) that warrant specific strategies for toxin neutralization (e.g. application of IVIG). Therefore, a host-derived, easy-to-measure signature that rapidly differentiates between streptococcal and other types of NSTIs seems to be of high clinical importance. Moreover, our results highlight the potential of host derived biomarkers for microbial diagnosis in NSTIs and hopefully will spark further investigations into their potential for improving treatment of this devastating disease.

5- Please justify and clarify the definition of mono- and polymicrobial NSTIs (p.6 line 108). Is this the standard definition? I may be missing something here but what happens if Gini-Simpson diversity is between 0.25 and 0.5?

Answer: The standard definitions (types) of NSTIs are typically based on culturing results. NSTI is considered "mono-infection" when only one pathogen can be isolated whereas polymicrobial NSTIs are often defined as "a mix of aerobic and anaerobic bacteria" (see for example ref 7 Cocanour et al or ref. 8 Elliott et al.). As it is well documented that some microorganisms are difficult or even impossible to culture with the current techniques, such a strict definition is difficult to hold in light of the increased sensitivity of culture-independent sequencing methods for microbial diagnosis. Using sensitive culture-independent methods and sufficient sequencing depth, there will never be a 100% "pure" mono-infection (e.g. due to "contaminating typical skin inhabitants"), causing traditional, culture-based classification schemes to be impossible to apply to sequencing-based results. Hence, we use the terminologies "monomicrobial" and "polymicrobial" as purely descriptive terms. While there is no "standard definition" for the sequencing data based classification of NSTIs, we consider in this study "mono-infection" as an infection that is dominated by a single pathogen, whereas "polymicrobial infections" are characterized by the presence of a set of microbes with a "significant" relative abundance.

Microbial ecology has developed a variety of diversity indices to describe whether a community is dominated by one single organism or is composed of several species with different abundance. One of the most commonly used indices is the Gini-Simpson index ($1-\lambda$), which is a measure of diversity, which takes into account both richness and evenness. The value ranges between 0 and 1 where 1 represents infinite diversity and 0 no diversity. As we use the terms 'polymicrobial' and 'monomicrobial' on a purely descriptive basis to describe the composition of the bacterial communities associated with NSTIs we decided to refer to cases where a single phylotype comprised >85% of the community as a mono-infections. This included all cases with bacterial communities with a Gini-Simpson index <0.25. Due to the descriptive nature of the used terminology, we have eliminated the reference to an index >0.5 when referring to "polymicrobial" infections as it could be falsely understood as an accepted threshold for infection-classification.

To delineate different types of NSTIs based on their bacterial etiology we later use bacterial co-occurrence modeling, and sample-wise hierarchical clustering to define different "pathotypes" (see Supplementary Figure 8, Figure 2). We thought that such a statistics-based approach should represent the most "unbiased" way of defining different "types" of NSTIs for the purpose of comparative analysis. We modified the language in the result section of the 16S data analysis to clarify this point.

6- Please point to the methods in p7 line 131 to explain what divisive clustering of co-occurring genera means. How robust is this clustering to other clustering algorithms?

Answer: We have included a statement in the revised manuscript that directs the reader to the method section for more details on the clustering methods used in this study. During data assessment we evaluated several algorithms commonly used to analyze network topology and identify highly interconnected modules. While the affiliation of low abundant taxa to the

identified modules varied slightly between algorithms (e.g. Girvan-Newman fast greedy algorithm, MCODE algorithm), highly abundant taxa were consistently grouped together. We decided to use the Markov clustering (MCL) algorithm as it has been successfully applied to a broad spectrum of biological data (microbiome data, gene networks, protein-protein interaction networks) and frequently outperforms other clustering algorithms (see e.g. Saelens et al Nat Comm, 2018, PMID: 29545622). We included a similar sentence in the Method section to clarify the reasons for our decision.

7- Please be more specific in the caption of Fig. 1c (e.g., say what dots represent)

Answer: We updated the legend to Fig 1c accordingly.

8- line 137, Fig 1c Please provide the corresponding statistic to justify significance.

Answer: We now added the information that "Diversity indices across the five body parts (n=14, 18, 12, 23 and 37, respectively from left to right) were compared using the Kruskal–Wallis with Dunn’s multiple comparison post-hoc test". We also added a sentence to the legend of Fig. 1c to indicate the statistics used to assess inter-bodypart differences of bacterial community composition: 'For statistical evaluation of the relative abundance of genera at different body sites see Table 1'.

9- Several figures only show the average values (mean). Is this representative of the underlying population? It would be more informative to show the standard deviation or confidence intervals with respect to the mean. This may be hard for some figures e.g., stacked barplots but it would certainly give a better understanding of the corresponding data.

Answer: In general, we aimed to indicate or directly depict inter-sample variability wherever possible (see e.g. Figures 1a, 3, 4b, 5b or 6, Supplementary Figures 9c, 10a) as we agree with the reviewer that it is an important feature of the dataset and necessary for accessing the validity of the authors conclusions. The reviewer is right that adding SD or CI to a stacked barplot can make plots hard to interpret. We therefore provide all the raw data necessary to access the validity of our conclusions and reproduce our analysis as supplementary tables in our submitted manuscript. We additionally updated figures that did not depict sample specific data used for generating averages depicted (Fig. 4a, c) to include an indication of the underlying variability or explicitly pointed readers towards the statistics used to assess the differences between compared groups (Fig. 1c – "For a statistical evaluation of the relative abundance of genera at different body sites see Table 1"). We additionally removed lowly expressed IPR domains (expressed at $<1 \log_2$ TPM by either pathobiome) from Fig. 4a to increase figure readability.

10- Fig2b: PCoA -> Principal Coordinates Analysis (PCoA)

Answer: The figure and the legend to the figure have been revised accordingly.

11- line 151: shouldn't this be just microbial community composition? If not then please provide more details.

Answer: That has been corrected accordingly.

12- line 154: "we randomly selected". Please point to the supplementary table and also state how many.

Answer: The numbers are now given in the main text, and the previous Supplementary Table 6 (now Table S4) has been upgraded to contain more comprehensive information.

13- I understand why only 17 monomicrobial NSTIS were chosen but one of my main concerns was that only 22 polymicrobial NSTIs were analyzed. Please explain if there is a particular reason behind this choice.

Answer: Our goal was to perform transcriptional profiling on more polymicrobial samples than those included in the current analysis. However, it has to be noted that the biopsies we received from surgical interventions at hospital submission were collected at random locations of the affected tissue. This frequently resulted in low bacterial abundances in the specific available biopsy and, consequently, low relative proportions of bacterial reads in relation to host reads in the generated RNAseq data. We sequenced additional samples which are not included in the manuscript because they did not yield sufficient proportions of bacterial (or human reads) and had to be excluded as they did not allow proper statistical analysis. An additional frequent limitation was poor quality of the extracted RNA (RNA integrity) from necrotic tissue (we currently include 22/45 biopsies from patients with polymicrobial NSTIs – see upgraded Supplementary Table 4). We included a sentence in the Method section to clarify this problem and the resulting exclusion criteria for individual samples. Overall, we think that we have generated the best possible dataset based on the available biological material.

14- line 156: Fig2. Is this the correct reference/figure?

Answer: Figure 2 shows the classification of all biopsies from where subsets of the most abundant groups were selected for sequencing. To give the reader a more comprehensive overview we have now added additional information to Supplementary Table 4 and reference it at the line in question.

15- line 163, line 108: This may be a naive question: Is there a reason why two measures of diversity are used in the paper? (alpha-diversity, Gini-Simpson diversity index).

Answer: Alpha diversity is generally the species richness and diversity within one ecosystem whereas for example beta diversity is the difference in diversity of species between two or more ecosystems in an area. Various indices have been developed to describe "alpha diversity", where the Gini-Simpson index is one of the most commonly used and takes into account the number of species present, as well as the abundance of each species. We are, thus, just presenting alpha diversity as described by the Gini-Simpson index. We added a sentence to the Materials section to clarify this point and upgraded the results section to make this point clearer.

16- line 168-178: Better annotation is needed. E.g., I couldn't find oligosaccharide core region biosynthesis, lipid A biosynthesis, polysaccharide transport or LPS annotated in Fig 3 or Supp Table 7.

Answer: Figure 3 gives a high-level overview about the GO terms associated with genes highly expressed by the bacterial communities in polymicrobial and streptococcal NSTIs. The GO-terms depicted are given on top of the graph using their common nomenclature (e.g.

GO:0009245 - lipid A biosynthetic process). We agree with the reviewer that the absence of individual labels highlighting functionalities discussed in the text on the bottom of the graph made it difficult for the reader to assess specific results highlighted in the text. We upgraded the figure to enable the reader to identify the features described within the result section more easily. We additionally added referenced GO terms to the main text or referred to the figure depicting KEGG pathways (Supplementary Figure 10) were appropriate. Also, the reference in the main text was referencing the wrong Supplementary Table. This has been corrected and all references to supplementary material in the main text have been revised.

17- line 268: *How did you calculate 85% accuracy? Figure 7f shows 21 black dots that were correctly classified (predicted probabilities >0.5) out of 27. This gives 21/27 = ~77% accuracy. Overall this entire section should be revised and more rigorous analysis/evaluation of the classifiers should be performed (see my initial comments).*

Answer: We have revised the entire section as recommended by the reviewer. Regarding the predicted probability, we have included a confusion matrix for evaluating the performance classification of the model (Fig. 7d). In the confusion matrix, the number of samples correctly and incorrectly classified by the model (predicted class) are compared with the actual class. The classification error for each class is also included in the confusion matrix. The model was able to correctly classify 23 of 32 non-streptococcal NSTIs patients and 23 of 27 *S. pyogenes* NSTIs patients.

Reviewer #2

(Remarks to the Author): This is a timely and well designed study that will provide valuable information for clinicians and researchers. The methodology is State of the Art. These are of course devastating infections and the authors suggest a 70% mortality. This is probably much higher than other studies they cite.

Answer: The reviewer is right to point out that the previous phrasing could be misleading to the reader. We edited the respective section of the manuscript to clarify this point.

Still without proper diagnosis and emergent intervention, that may be true. Few publications have had such a comprehensive evaluation of the microbes and potential pathogenic mechanisms. The authors are to be commended. The authors have defined their studies according to a zero time and this is the point they obtained specimens from blood and operative procedures. I can understand their definition. Yet numerous studies, which they cite, all suggest that the time to surgery is critical in determining outcome. What was the mortality in this study? What was the time from onset of signs and symptoms to operative procedures? They define this as zero time. I do not think that is sufficient.

Answer: The reviewer is correct in pointing out that the duration from onset of symptoms until initial surgery is a critical aspect for disease outcome. However, this timeframe is hard to determine accurately as it is often not included in the patient files. Our study was, however, designed to identify host signatures that could be used to robustly differentiate patients with streptococcal and polymicrobial NSTIs irrespective of the time since onset. As our study design was agnostic to variability in the time since onset, we would actually argue that, assuming an underlying variability in this timeframe, our approach would only identify signatures that are more robust to time-dependent fluctuations. We included a sentence in the introduction that aims to clarify this point. We, however, agree with the reviewer that mortality is important to contextualize the impact of the differences in pathophysiology that we observed between the different types of NSTI. Therefore, we have added a

Supplementary Figure depicting the mortality of the group of patients included in our study from whom 16S sequencing data are available (Supplementary Figure 8b).

What for the purpose of the study was their definition of a necrotizing infection? Was this based on histopathology? If so what were those findings. Surgeons impression, gram stains, evidence of septic shock or path reports? These should be included in the criteria for entrance into the study.

Answer: We have updated the method section (study design) and referenced the study protocol recently published (Madsen MB, Skrede S, Bruun T, et al. Necrotizing soft tissue infections - a multicentre, prospective observational study (INFECT): protocol and statistical analysis plan. *Acta Anaesthesiol Scand* 2018; 62: 272–9.) to more clearly define how NSTIs were diagnosed for the purpose of this study.

The include a reference to a study by Olsen and Musser that states that proteolytic activity in group A streptococcal necrotizing infection was important. Yet their results describe proteolytic activity as being more representative of poly-microbial necrotizing infections.

Answer: The reviewer is right to point out this apparent inconsistency. The study by Olsen and Musser does, however, solely describes the known mechanisms of streptococcal virulence that are of importance during NSTIs, without contrasting it with other 'types' of NSTIs such as NSTIs of polymicrobial etiology. While our results show that the extracellular proteolytic activity may be higher during polymicrobial infections we also show that several genes encoding factors (Sic, M-protein, Streptolysin), including extracellular proteases (ScpA, ScpB, SpeB), described in the manuscript by Olsen and Musser are highly expressed by streptococcal pathogens during NSTI (see Figure 4 a,b). Therefore, we do not think that our results contradict the results of this prior study, but rather build upon and expand the description of the molecular pathogenesis of NSTIs.

Those data that suggest that interferon mediators allow, differentiation between polymicrobial and group A streptococcal infection may be statistically significant, there is great overlap in quantitative results. So in a given patient this may not be diagnostically accurate, especially if one considers the time of onset of infection, and they only provide one data point which they define as zero time.

Answer: We agree with the reviewer that for individual patients the identified signature is insufficient to be used as the sole method of microbial diagnosis yet. We, however, think that diagnostic tools based on fast identification of host-derived biomarker would drastically reduce the turnaround time when compared to culture-based methods for microbial diagnosis. Moreover, the identified signature is seemingly relatively robust to variation in time since onset (assuming an underlying variation in the patient cohort). We have also included a new paragraph in the Discussion section specifying that our study is exploratory and need to be further validated using larger samples set.

I consider these important issues that should be addressed.

Reviewers' comments:

Reviewer #1 (Remarks to the Author):

I would like to thank the authors for addressing my comments in their revised manuscript. I am overall satisfied with the point-to-point response and I really appreciate the authors' effort in clarifying the corresponding parts of the paper. However my concern regarding the classifier performance evaluation still remains (comment #17).

At the very least the authors should clearly state in the main text the error rate of the classifier and include the corresponding ROC. The error rate is equal to the total number of errors (False Positives + False Negatives) divided by the total number of cases (n) where False Positives = 'Other' incorrectly classified as Streptococcal, False Negatives = 'Streptococcal' incorrectly classified as 'Other'.

According to the confusion matrix (new figure 7d), the current error rate is $(11+4)/61 \approx 25\%$ which is unacceptably high for a diagnostic test. However it may be the case that False Positives are more important than False Negatives in a clinical setting. A discussion on this, together with the validation ROC would be helpful in choosing the right operating point for this classifier.

A more important concern however is that I still find a few discrepancies in the reported data. In my initial review I had asked for example how the reported 85% accuracy was calculated since I had trouble validating the results. Unfortunately the same goes for the current version of the manuscript:

The authors state that they validated the classifier in an independent cohort with 59 cases. On the other hand, the corresponding confusion matrix reported in Figure 7d shows $23+23+4+11=61$ cases. The Supp. Table 4 shows only 58 cases that were included in the validation cohort (marked with E). In the supplementary data for Supp. Fig 13 there are 59 cases. The one extra sample number (that is not marked with an 'E' in Supp. Table 4) seems to be 2133. In the supplementary data this sample is labeled as 'other'. In Supp. Table 4 however this sample seems to correspond to a Streptococcal infection.

These errors/inconsistencies make it very hard to trust the author's quantitative analysis in this work and I am now wondering whether this affects other parts of the paper as well. For this paper to be accepted for publication the authors should go over their results very carefully and cross validate for consistency. One way that the authors can ensure the quality of the reported data would be to provide the code/scripts used to generate the data/figures as a supplementary file so that I (and perhaps the other reviewers as well) can have a step by step reference on the specific parts of the analysis.

Reviewer #2 (Remarks to the Author)

While the authors reference a publication, they did not address the clinical criteria of necrotizing infections that they used to identify patients to be included. As stated before, there are well established criteria ranging from clinical presentation, heart rate, blood pressure, renal status, WBC, CRP, histopathology, surgeons descriptions of the findings at debridement, gram stains, culture results. These are critically important to put their study in the perspective that clinicians can identify with.

We could all agree that the time to surgery from presentation is a critical decision process. Probably greater than 50% of the articles written by surgeons relative to better outcomes focus on the time to surgical debridement.

This article could be a milestone in providing modern technology to shorten this time frame. Without a clear picture of their patients presentations, this will be a failure to improve patient care.

HOW LONG DID THEIR PATIENTS HAVE SYMPTOMS COMPATIBLE WITH A NECROTISING INFECTION BEFORE THEY HAD THEIR STUDY INTERVENTIONS?

Response to the reviewer's comments:

Reviewer #1

(Remarks to the Author): I would like to thank the authors for addressing my comments in their revised manuscript. I am overall satisfied with the point-to-point response and I really appreciate the authors' effort in clarifying the corresponding parts of the paper. However my concern regarding the classifier performance evaluation still remains (comment #17). At the very least the authors should clearly state in the main text the error rate of the classifier and include the corresponding ROC. The error rate is equal to the total number of errors (False Positives + False Negatives) divided by the total number of cases (n) where False Positives = 'Other' incorrectly classified as Streptococcal, False Negatives = 'Streptococcal' incorrectly classified as 'Other'. According to the confusion matrix (new figure 7d), the current error rate is (11+4)/61 ≈ 25% which is unacceptably high for a diagnostic test. However, it may be the case that False Positives are more important than False Negatives in a clinical setting. A discussion on this, together with the validation ROC would be helpful in choosing the right operating point for this classifier.

Answer: As requested by the reviewer, we have included an additional panel in Figure 7 showing the final ROC.

Taking the correct numbers (see comments below), 23 of 27 *S. pyogenes* NSTIs could be correctly classified, corresponding to a true positive rate of 85.2%. Accordingly, we had stated in the first version of our manuscript on "85% correctly classified *S. pyogenes* NSTIs". We never reported 85% as "accuracy", which is (due to the higher error in classifying non-*S. pyogenes* NSTIs) $(21+23) / (21+11+23+4) = 0.746$. We upgraded the text accordingly and included the requested information.

We agree with the reviewer that the error rate of our classifier is too high to be of immediate diagnostic value in the clinical setting. However, there are two main reasons why we think that this part of our manuscript is of direct medical importance.

First, detection of *S. pyogenes* NSTIs is of the utmost clinical importance, as these infections are frequently associated with high systemic toxicity (see Stevens et al. *NEJM* 2017). For patients suffering from *S. pyogenes* NSTIs, prompt surgical exploration is critical, as timely debridement, amputation and rapid containment of systemic toxicity are often required to curb progression of the infection. Our classifier shows promising capabilities to identify particularly this type of NSTIs, illustrating that readily assessable host-derived signatures may be particularly useful to identify the type of infection that requires rapid intervention.

Second, we believe that this part of our manuscript has a largely exploratory character, which we state in the main text and which was appreciated by Reviewer #1 in the first round of comments to our manuscript. We believe that our classifier shows that readily quantifiable host-derived signatures can assist rapid microbial diagnosis. Moreover, we lay the foundation for future research for a set of biomarkers that is able to reliably distinguish between NSTI types with clinically desirable accuracy.

In concordance with the reviewer's suggestion, we have added a sentence to the result section and edited the relevant section of the discussion to more clearly highlight those two points.

A more important concern however is that I still find a few discrepancies in the reported data. In my initial review I had asked for example how the reported 85% accuracy was calculated since I had trouble validating the results. Unfortunately, the same goes for the current version of the manuscript: The authors state that they validated the classifier in an independent cohort with 59 cases. On the other hand, the corresponding confusion matrix reported in Figure 7d shows 23+23+4+11=61 cases. The Supp. Table 4 shows only 58 cases that were included in the validation cohort (marked with E). In the supplementary data for Supp. Fig 13 there are 59 cases. The one extra sample number (that is not marked with an 'E' in Supp. Table 4) seems to be 2133. In the supplementary data this sample is labeled as 'other'. In Supp. Table 4 however this sample seems to correspond to a Streptococcal infection.

Answer: We apologize for the confusion. To clarify: The validation cohort comprised 59 patients as given in the source data for supplementary Fig. 13. Accordingly, we had stated in the text: "The potential of these plasma biomarkers for the identification of *S. pyogenes* NSTIs was then tested in an independent patient cohort (n = 59), of which 27 were identified by 16S rRNA gene sequencing as monomicrobial *S. pyogenes* NSTIs and 32 as NSTIs caused by other microorganism(s) (Supplementary Table 4)". The reviewer is right that the labeling of case 2133 as 'E' in table 4 was missing. This has been corrected. Similarly, the reviewer is right that figure 7d included a mistake. As stated above (and as shown in the source data file), the validation cohort comprised 59 cases of which 27 were *S. pyogenes* NSTIs 32 were NSTIs caused by other microorganisms. Our goal in this part of the manuscript was to specifically train a classifier that can detect *S. pyogenes* NSTIs based on a limited set of plasma biomarkers. This goal is clearly stated in the subsection headline. Therefore, case 2133, dominated by *S. dysgalactiae* as well as cases 3049 and 2156 dominated by *S. agalactiae* or *S. constellatus*, respectively, were grouped as 'non-*S. pyogenes*'. As we now realize the previously used nomenclature ('Others') for these cases could have caused confusion of the reader. We therefore adjusted the nomenclature in the whole section to '*S. pyogenes* NSTIs' and 'non-*S. pyogenes* NSTIs', to avoid any overlap with previously used terminology. The source data file has been updated to include the correct nomenclature for Supplementary Figure 13 data.

These errors/inconsistencies make it very hard to trust the author's quantitative analysis in this work and I am now wondering whether this affects other parts of the paper as well. For this paper to be accepted for publication the authors should go over their results very carefully and cross validate for consistency. One way that the authors can ensure the quality of the reported data would be to provide the code/scripts used to generate the data/figures as a supplementary file so that I (and perhaps the other reviewers as well) can have a step by step reference on the specific parts of the analysis.

Answer: We apologize for the unclarities. In line with the reviewer's suggestions, we have added access to relevant code/scripts of our manuscript (https://github.com/MoIProfileStrepNSTI/NSTI_src_code) to enable step-by-step assessment of the validity of our conclusions. We, moreover, provided all relevant source data underlying all figures and graphs. We hope that this facilitates in-depth assessment of the validity of our conclusions.

Reviewer #2

(Remarks to the Author): While the authors reference a publication, they did not address the clinical criteria of necrotizing infections that they used to identify patients to be included. As stated before, there are well established criteria ranging from clinical presentation, heart rate, blood pressure, renal status, WBC, CRP, histopathology, surgeons descriptions of the findings at debridement, gram stains, culture results. These are critically important to put their study in the perspective that clinicians can identify with.

Answer: In our previous revised version of the manuscript we had stated that "Diagnosis of NSTIs was performed by the acting surgeon at primary operation or revision as described earlier⁴⁸. Briefly, diagnosis was based on the presence of necrotic or deliquescent soft tissue with widespread undermining of the surrounding tissue".

We agree with the reviewer that the referenced values (white blood cell count, c-reactive protein levels, hemoglobin concentrations, serum sodium, creatinine or glucose levels – often compiled in the LRINEC score) have previously shown value for delineating NSTIs from different clinical entities like cellulitis, erysipelas, and cutaneous abscesses. However, neither of these values nor LRINEC scores are used as definite diagnostic criteria and recent literature has called their diagnostic value for NSTIs into question (e.g., Neeki et al. *Western Journal of Emergency Medicine* 2017). Importantly, the gold-standard for clinical diagnosis of NSTIs is “via surgical exploration of the soft tissues in the operating room, with physical examination of the skin, subcutaneous tissue, fascial planes, and muscle” (Stevens et al. *UpToDate* Dec 2018, see also Stevens et al *NEJM* 2017). Consequently, this procedure was used to establish ultimate diagnosis, as referenced in the method section of the manuscript (see reference #51).

However, we agree with the reviewer that these values could help clinicians to put our studies results into perspective. Therefore, we now provide extensive laboratory values (concentrations of haemoglobin, C-reactive protein, creatinine, sodium, glucose, and white blood cell counts), as well as physiological values (pulse rate and mean arterial pressure) and LRINEC scores for all identified infection types in the supplement of our manuscript (Supplementary Table 12).

We could all agree that the time to surgery from presentation is a critical decision process. Probably greater than 50% of the articles written by surgeons relative to better outcomes focus on the time to surgical debridement. This article could be a milestone in providing modern technology to shorten this time frame. Without a clear picture of their patients presentations, this will be a failure to improve patient care. HOW LONG DID THEIR PATIENTS HAVE SYMPTOMS COMPATIBLE WITH A NECROTISING INFECTION BEFORE THEY HAD THEIR STUDY INTERVENTIONS?

Answer: We agree with the reviewer that shortening the timeframe from diagnosis to surgical intervention is critical to improve patient care in this rapidly progressing infection. However, the time of ‘onset of symptoms’ is often ambiguous and could be understood as either the time at which an individual first experienced any symptoms of illness, time of first doctor visit, or the time when a treating physician first notes potential symptoms associated with NSTI. Previous literature fails to establish a clear definition but frequently uses the terminology ‘delayed diagnosis is used’, highlighting the issue of diagnostic uncertainty associated with NSTIs. Because of this ambiguity in defining the onset of NSTI we did not

demarcate a 'time of onset' in this study. A section has been added to the discussion highlighting this problem.

As we, however, agree with the immense value of shortening diagnostics timeframes for NSTIs we have added available information on the time from hospital admission to surgical intervention for all patients of all infection types (Supplementary Table 12). We think that this parameter is most relevant for supporting our study results, as we specifically focused on identifying signatures that allow rapid microbial diagnosis at hospital admission, to optimize time until intervention.

REVIEWERS' COMMENTS:

Reviewer #1 (Remarks to the Author):

I would like to thank the authors again for taking the time to go over my suggestions and address my concerns regarding the classifier performance and the reported statistics. The manuscript now highlights the exploratory nature of the corresponding results and the potential limitations of the proposed classifier which in my opinion puts the authors' contributions in the right context. I also really appreciate their effort in providing the source code to support their quantitative analysis. Overall the authors have addressed all of my comments from the previous round of reviews and as far as I can tell the reported results are now consistent throughout the manuscript.

Reviewer #2 (Remarks to the Author):

The authors state that routine lab tests suggested by me are to be found in supplemental table 12. This I could not find. The authors have added a couple of words that necrotizing infection was diagnosed by necrosis and undermining of tissue according to the surgeons expertise. Surgeons generally have more definitive definition of such infections and to me the author's responses are not responsive. In addition the authors state that time 0 was when they started their study, while this is convenient for their purpose it does not give anyone a sense of how long patients had been ill, had they been surgerized before transfer to their research hospital, etc. The whole premise of their study is to demonstrate that potentially certain biomarkers could predict group A streptococcal infection and that this could reduce time to surgery and improve outcome. This paper could go along way to help in that respect but the clinical information which describes the severity and timing of the infection is crucial in my opinion and that has not thus far been answered by their responses. While they state it is difficult to determine how long their patients have been ill, that is not in my opinion acceptable. They should also report in this series, the time it took surgeons to operate and what the mortality and morbidity were. The day 0 designation is really misleading.

The abstract says that these results are unprecedented may be a little strong.

This work is important but would be strengthened by some markers of clinical relevance. Truly they should be commended by doing this type of molecular biomarker work on these devastating infections. If they cannot supply the clinical information perhaps it could be drastically reduced to a brief report.

Response to the reviewer's comments:

Reviewer #1

I would like to thank the authors again for taking the time to go over my suggestions and address my concerns regarding the classifier performance and the reported statistics. The manuscript now highlights the exploratory nature of the corresponding results and the potential limitations of the proposed classifier which in my opinion puts the authors' contributions in the right context. I also really appreciate their effort in providing the source code to support their quantitative analysis. Overall the authors have addressed all of my comments from the previous round of reviews and as far as I can tell the reported results are now consistent throughout the manuscript.

Answer: We would like to thank the reviewer for the fair and constructive comments throughout the review process of this manuscript, which helped to improve the quality of our work.

Reviewer #2

The authors state that routine lab tests suggested by me are to be found in supplemental table 12. This I could not find.

Answer: The supplementary table 12 could be found in the supplementary information under the heading "supplementary tables" but was obviously missed by the reviewer. As suggested by the editor these data are now moved to the main text (new Table 1).

The authors have added a couple of words that necrotizing infection was diagnosed by necrosis and undermining of tissue according to the surgeons expertise. Surgeons generally have more definitive definition of such infections and to me the author's responses are not responsive.

Answer: We provided the most important values guiding clinical diagnosis when a NSTI is suspected in previous supplementary Table 12. These data are now to be found in Table 1. However, ultimate diagnosis can only be established by surgical exploration of affected tissue. Therefore, our study has a *a priori* set its diagnostic criteria as previously stated: "Diagnosis of NSTIs was performed by the acting surgeon at primary operation or revision as described earlier⁴⁸. Briefly, diagnosis was based on the presence of necrotic or deliquescent soft tissue with widespread undermining of the surrounding tissue".

In addition, the authors state that time 0 was when they started their study, while this is convenient for their purpose it does not give anyone a sense of how long patients had been ill, had they been surgerized before transfer to their research hospital, etc. The whole premise of their study is to demonstrate that potentially certain biomarkers could predict group A streptococcal infection and that this could reduce time to surgery and improve outcome.

Answer: We again, want to point out the complications with establishing a robust definition of 'onset', which we brought up during the previous round of revision. 'Onset' could be

defined as the time at which an individual first experienced any symptoms of illness, time of first doctor visit, or the time when a treating physician first notes potential symptoms associated with NSTI. Previous literature and medical theory fail to provide a clear definition. These complications prevented us from being able to provide a reliable timeframe for 'onset' to hospital admission for our patient cohort.

However, the purpose of this study was not primarily to optimize time to surgery but "to identify the bacterial taxa associated with NSTIs, identify their functionalities that contribute to disease pathophysiology and identify host signatures of potential value for rapid microbial diagnosis at the time of hospital admission." We hope that the insights presented in this study can help to rapidly adjust medical treatment (i.e. antibiotic, adjunct therapy), by facilitating rapid microbiological diagnosis.

This paper could go along way to help in that respect but the clinical information which describes the severity and timing of the infection is crucial in my opinion and that has not thus far been answered by their responses. While they state it is difficult to determine how long their patients have been ill, that is not in my opinion acceptable. They should also report in this series, the time it took surgeons to operate and what the mortality and morbidity were. The day 0 designation is really misleading.

Answer: We have provided additional clinical information on our patient cohort in Table 1 (previous supplementary Table that was missed by the reviewer) to provide more context on our patient cohort. This data includes time-resolved mortality information, which should help to judge the relative severity of the disease per infection type.

As pointed out above, we can not provide information for the time individual patients have been affected by NSTI as this study did not define a point of 'onset' for patients. We have additionally provided detailed per-infection-type information for the time from hospital admission to surgical intervention which should help to contextualize our analysis.

The abstract says that these results are unprecedented may be a little strong.

Answer: We have adjusted the phrasing in the abstract, according to the reviewer's suggestion.

This work is important but would be strengthened by some markers of clinical relevance. Truly they should be commended by doing this type of molecular biomarker work on these devastating infections. If they cannot supply the clinical information perhaps it could be drastically reduced to a brief report.

Answer: We have added substantial patient metadata to our study (Table 1) and hope that this additional information helps to contextualize our study for medical practitioners. We hope that our results will help spark additional research to validate and expand on our panel of potential biomarkers, as new treatment strategies are desperately needed for this still devastating disease.